# Unveiling spatial and temporal heterogeneity of a tropical forest canopy using high-resolution NIRv, FCVI, and NIRvrad from UAS observations

Trina Merrick[1], Stephanie Pau[2], Matteo Detto[3,4], Eben N. Broadbent[5], Stephanie. A. Bohlman[3,6], Christopher J. Still[7], Angelica M. Almeyda Zambrano[8]

[1] Naval Research Laboratory, Remote Sensing Division, 4555 Overlook Ave. SW, Washington, DC,20375, USA
[2] Department of Geography, Florida State University, 113 Collegiate Loop, Tallahassee, Florida 32306, USA
[3] Smithsonian Tropical Research Institute, Apartado 0843–03092, Balboa, Ancon, Panama
[4] Department of Ecology and Evolutionary Biology, Princeton University, Princeton, New Jersey 08544 USA
[5] Spatial Ecology and Conservation Lab, School of Forest, Fisheries and Geomatics Sciences, University of Florida, Gainesville, FL, 32608 USA
[6] School of Forest, Fisheries and Geomatics Sciences, University of Florida, Gainesville, FL, 32608 USA
[7] Department of Forest Ecosystems and Society, Oregon State University, Corvallis, Oregon 97331 USA
[8] Spatial Ecology and Conservation Lab, Center for Latin American Studies, University of Florida, Gainesville, Florida 32608 USA
[*2] Primary author former affiliation

*Correspondence to*: Trina Merrick (trina.merrick@nrl.navy.mil)

**Abstract.** Recently, remotely-sensed measurements of the near-infrared reflectance (NIRv) of vegetation, the fluorescence correction vegetation index (FCVI), and radiance (NIRvrad) of vegetation, have emerged as indicators of vegetation structure and function with potential to enhance or improve upon commonly used indicators, such as the normalized difference vegetation index (NDVI) and the enhanced vegetation index (EVI). The applicability of these remotely sensed indices to tropical forests, key ecosystems for global carbon cycling and biodiversity, have been limited. In particular, fine-scale spatial and temporal heterogeneity of structure and physiology may contribute to variation in these indices and the properties that are presumed to be tracked by them, such as gross primary productivity (GPP) and absorbed photosynthetically active radiation (APAR). In this study, fine-scale (approx.15cm) tropical forest heterogeneity represented by NIRv, FCVI, and NIRvrad, and by lidar-derived height is investigated and compared to NIRV and EVI using unoccupied aerial system (UAS)-based hyperspectral and lidar sensors. By exploiting near-infrared signals, NIRv, FCVI, and NIRvrad captured the greatest spatiotemporal variability, followed by the enhanced vegetation index (EVI), then the normalized difference vegetation index (NDVI). Wavelet analyses showed the dominant spatial scale of variability of all indicators was driven by tree clusters and larger-than-tree-crown size gaps rather than individual tree crowns. NIRv, FCVI, NIRvrad, and EVI captured variability at smaller spatial scales (~50 m) than NDVI (~90 m) and lidar-based surface model (~70 m). We show that spatial and temporal patterns of NIRv and FCVI were virtually identical for a dense green canopy, confirming predictions in earlier studies. Furthermore, we show that NIRvrad, which does not require separate irradiance measurements, correlated more strongly with GPP and PAR than did other indicators. NIRv, FCVI, and NIRvrad, which are related to canopy structure and the radiation regime of vegetation canopies, are promising tools to improve understanding of tropical forest canopy structure and function.

## 1 Introduction

Important spatial and temporal heterogeneity in structurally complex and species-rich tropical forests is not well characterized. Many factors contributing to this heterogeneity, including varying microclimate, light conditions, topography, crown structure, and patterns of tree mortality and regeneration, can produce high variability in carbon fluxes, ultimately affecting coarse-scale gross primary production (GPP) measurements in forests (e.g., Xu et al., 2015; Guan et al., 2015; Morton et al., 2014; Bohlman and Pacala, 2012; Laurance et al., 2012; Clark et al., 2008; Huete et al., 2008). Improving knowledge of tropical forest dynamics at multiple scales is crucial to monitoring and predicting resilience of tropical ecosystems and productivity under climate change (Liu et al., 2021; Clark et al., 2017; Laurance et al., 2012; Malhi, 2012; Wright, 2010; Saatchi et al., 2010; Lewis et al., 2009). Remote sensing (RS) measurements have been employed to uncover vegetation patterns of structure and productivity from local to global scales, often with a focus on filling gaps in knowledge regarding variation and uncertainties in GPP estimates (e.g., Jung et al., 2011; Glenn et al., 2008; Huete et al., 2002; Ryu et al., 2018; Yang et al., 2017; Jiang et al., 2008; Zhao et al., 2010; Heinsch et al., 2006; Running et al., 2004; Turner et al., 2003). Yet, the spatial mismatch between satellite data (e.g., 30 m to 1 km pixel resolution), which provides observations across large extents at repeat intervals, and site-specific plot level data (e.g., 0.1 – 1 hectare), is in part responsible for the uncertainties in GPP estimates Yet, there is a spatial mismatch between satellite data (e.g., 30 m to 1 km pixel resolution), which provides observations across large extents at repeat intervals, and plot level data , is in part responsible for the uncertainties in GPP estimates (Gelybó et al., 2013; Zhang et al., 2020). A way to solve this problem is to acquire high spatial and temporal resolution data that can capture fine-grained heterogeneity of tropical forests (Clark et al., 2017; Mitchard, 2018; Saatchi et al., 2011; Lewis et al., 2009). Unoccupied aerial systems (UAS) with hyperspectral imaging sensors offer an opportunity to collect tropical forest canopy data at high spatial resolution and which could address unknowns related to the high heterogeneity of tropical forests.

Traditional reflectance-based indices (RI) using RS data, such as the normalized difference vegetation index (NDVI) and enhanced vegetation index (EVI), are known to capture structural changes that are coincident with changes in GPP. RIs have provided optical methods using RS to track GPP via the light use efficiency (LUE) model (J.L.Monteith, 1977; Yuan et al., 2014; B. E. Medlyn, 1998). In the most commonly used formulation of the LUE model for RS, GPP is

$$GPP = APAR \; x \; \varepsilon \tag{1}$$

where APAR is the absorbed photosynthetically active radiation and ($\varepsilon$) is the efficiency with which the target vegetation converts the radiation to carbon (Gamon, 2015; Yuan et al., 2014; Running et al., 2004). APAR is derived from

$$APAR = PAR \; x \; fPAR \tag{2}$$

where PAR is the incoming photosynthetically active radiation and fPAR is the fraction of absorbed PAR. RIs commonly used in the LUE model of GPP as well as direct proxies for GPP are NDVI and EVI, because of a strong relationship to fPAR (Springer et al., 2017; Morton et al., 2015; Gamon et al., 2015; Porcar-Castell et al., 2014; Glenn et al., 2008; Gao et al., 2007; Huete et al., 2002; Zarco-Tejada et al., 2013). NDVI and EVI are typically used as

proxies on seasonal timescales. When used to examine changes on shorter timescales, they have been multiplied by
photosynthetically active radiation (PAR) to account for changes in radiation (incoming, absorbed, and scattered)
which better align with GPP changes (Springer et al., 2017; Yuan et al., 2014). However, RIs alone have often not
shown enough sensitivity to capture more fine-scale or rapid changes in vegetation, such as those in tropical forests,
and questions linger about the ability to track green-up with RIs in evergreen regions (Liu et al., 2021; Yang et al.,
2018a; Lee et al., 2013; Xu et al., 2015; Morton et al., 2014; Samanta et al., 2010; Sims et al., 2008).
Recently, three emerging vegetation indicators have been shown to track with GPP more closely than traditional
RIs. These indicators are the near-infrared reflectance of vegetation (NIRv) (Badgley et al., 2017), the fluorescence
correction vegetation index (FCVI) (Yang et al., 2020) and the near-infrared radiance of vegetation (NIRvrad) (Wu et
al., 2020). Because they exploit additional information from the NIR region of the spectrum, NIRv, FCVI, and
NIRvrad do not saturate in dense canopies or suffer the same level of contamination from senesced vegetation and
soils as traditional RIs (Baldocchi et al., 2020; Badgley et al., 2017). Additionally, these indicators require only
moderate spectral resolution data and are similarly straightforward to measure and calculate as RIs, making them
accessible in a broad range of studies. Therefore, NIRv, FCVI, and NIRvrad could be employed as valuable indicators
of canopy structure and function (Badgley et al., 2019; Badgley et al., 2017; Dechant et al., 2020).
NIRv is the product of NDVI and the total near-infrared scene reflectance (NIR). NIRv from moderate
spectral resolution satellite imagery and field spectrometers has been shown to empirically track both measured and
modelled GPP globally, although with highest uncertainties in the tropics. The NIRv~GPP relationship holds at
monthly to seasonal timescales presumably due to co-incident changes in canopy phenology, light capture and
scattering, and GPP (Badgley et al., 2019; Badgley et al., 2017; Dechant et al., 2020). FCVI, derived from radiative
transfer theory rather than an empirical relationship, is calculated from RS data by subtracting the reflectance in the
NIR from the reflectance in the visible range (Yang et al., 2020). Yang et al. (2020) demonstrated that FCVI tracked
GPP and solar-induced fluorescence (SIF; a radiance-based indicator of GPP), by capturing structure and radiation
information from a vegetated canopy in field experiments with crops and in numerical experiments. Yet FCVI showed
differences from NIRv due to exposed soil within the vegetated study areas. In previous studies, FCVI and NIRv were
similar for dense green canopies where soils have less of an impact, but this has not yet been tested in the tropics
(Wang et al., 2020; Badgley et al., 2019; Dechant et al., 2020). The product of NDVI and the NIR radiance, called
NIRvrad, was proposed as a proxy for GPP on half-hourly and daily timescales. In contrast, NIRv and FCVI track
changes on longer timescales (Wu et al., 2020; Dechant et al., 2020; Baldocchi et al., 2020; Zeng et al., 2019). Because
the radiance of NIR accounts for incoming radiation at short timescales, NIRvrad has tracked GPP and SIF on half-
hourly and diurnal scales as well as seasonally in crops and, to a limited extent, natural grass and savanna ecosystems
(Dechant et al., 2020; Baldocchi et al., 2020; Zeng et al., 2019; Wu et al., 2020).
Readily available UAS-based hyperspectral sensors are capable of robust measurements of NIRv, FCVI, and
NIRvrad at ultra-high spatial scales, i.e. tens of centimeters or less. In this regard, UAS-based data have the potential
to improve our understanding of tropical forest structure and function over a range of scales that are poorly resolved
by other RS platforms. Here, we use high spatial resolution UAS measurements to characterize spatial and temporal
variation in a semi-deciduous tropical forest canopy during the dry season, and compare commonly used spectral

indices (NDVI and EVI) to newer vegetation indicators (NIRv, NIRvrad, and FCVI) by (i) examining correlations between GPP and vegetation indicators using mean values across the canopy throughout the day, (ii) evaluating the distribution of fine spatial resolution values (~15 cm) across the canopy and examining changes in this spatial variation throughout the course of two days, and finally (iii) identifying the dominant spatial scale driving variation across our 10 ha study region.

## 2 Materials and Methods

### 2.1 Study Area

Barro Colorado Island (BCI), Panama, is a 1560 ha island (approximately 15 km$^2$) in Gatun Lake, which was formed by the construction of the Panama Canal. The Smithsonian Tropical Research Institute manages the preserved area specifically for research. This semi-deciduous moist tropical forest receives approximately 2640 mm mean annual precipitation and has a mean temperature of 26ºC with a dry season from approximately January through April (Detto et al., 2018). There is high species diversity, with approximately 500 tree species, approximately 60 species per ha, and about 6.3% of trees at >30cm diameter at breast height (dbh) (Bohlman and O'Brien, 2006; Condit et al., 2000). The UAS and ground measurements were focused on an area approximately 10 ha within the footprint of an eddy covariance tower near the center of the island (9.156440°, -79.848210°).

### 2.2 Data collection

The GatorEye Unmanned Flying Laboratory is a hardware and software system built for sensor fusion applications, and which includes hyperspectral, thermal, and visual cameras and a Lidar sensor, coupled with a differential GNSS, internal hard drives, computing systems, and an Inertial Motion Unit (IMU). Hardware and processing details, as well as data downloads, are available at www.gatoreye.org. The GatorEye flew 13 missions on January 30 and 31, 2019 over the forest canopy within the eddy covariance tower footprint at an average height of 120 m above ground level (AGL) and at 12 m/s (Fig. 1). In this study, we used radiometrically calibrated flight transects from the Nano VNIR 270 spectral band hyperspectral sensor (Headwall Photonics, Fitchburg, MA, USA) which covered approximately 1 ha per flight within the EC footprint in this study. The Nano sensor spectrally samples at approximately 2.2 nm and 12-bit radiometric resolution from 400 to 1050 nm. The frame rate was set to 100 fps, with an integration time of 12 ms and provided a pixel resolution of approximately 15x15 cm. The Nano was calibrated to radiance by the manufacturer before the field campaign and pixel drift was removed by dark images collection, which was corrected for during the conversion from digital number to radiance. The hyperspectral transects were equally subset for each flight in ENVI + IDL (Harris Geospatial, Boulder, CO). Each flight resulted in 1920 transects of approximately 400 m length composing three blocks discretized in 2500 data points. Simultaneous lidar was collected using a VLP-32c ultra puck (Velodyne, San Jose, CA), which was processed to a 0.5x0.5 m resolution digital surface model (DSM).

Turbulent fluxes and meteorological variables were measured from a 40 m Eddy Covariance (EC) flux tower (Fig. 1). The eddy covariance system includes a sonic anemometer (CSAT3, Campbell Scientific, Logan, UT) and an

open-path infrared CO2/H2O gas analyzer (LI7500, LiCOR. Lincoln, NE). High-frequency (10Hz) measurements
were acquired by a datalogger (CR1000, Campbell Scientific) and stored on a local PC. Other measurements made at
the tower include air temperature and relative humidity (HC2S3, Rotronic, Hauppauge New York), and
photosynthetically active radiation (PAR; BF5, Delta-T Devices, UK). EC data were processed with a custom program
using a standard routine described in Detto et al. (2010). GPP was derived from daytime values of net ecosystem
exchange (NEE) by adding the corresponding mean daily ecosystem respiration obtained as the intercept of the light
response curve (Lasslop et al., 2010). Due to a power issue, EC data were not available during the January 30 flights;
so only January 31 GPP were available.

156       An HH2 Pro Spectroradiometer (HH2; ASD/Panalytical/Malvern, Boulder, CO) fitted with a diffuse cosine

receptor was used on the ground in full sun at the forest edge to record incoming irradiance on January 30 and 31,
2019 (~ 3nm FWHM and spectral sampling at 1nm). HH2 irradiance was resampled to match the Nano hyperspectral
sensor and used to calculate reflectance. A calibrated reference tarp was placed in full sun at the forest edge and the
UAS flew over and recorded the tarp each UAS flight. Reflectance was calculated separately using the HH2 and tarp
data and resulting reflectance values compared as a method to vicariously cross-calibrate reflectance from the
hyperspectral data (<7.0% difference for all data in the study). In addition, PAR was calculated with the HH2 data and
compared to the tower-mounted PAR measurement (approximately 1.5 km apart) to help understand any differences
in the sky conditions during flight times. PAR differences across the site for each flight time for the duration of flights
(approximately 10-15 minutes in length each) ranged between 4.0% and 10.3%.

## 2.3 Vegetation indicators

We calculated NDVI and EVI as (Tucker, 1979; Huete et al., 2002; Rouse JR et al., 1974):

$$NDVI = \frac{R_{770-800} - R_{630-670}}{R_{770-800} + R_{630-670}} \tag{1}$$

and

$$EVI = \frac{2.5(R_{770-800} - R_{630-670})}{R_{770-800} + 6 \times R_{630-670} - 6 \times R_{460-475} + 1} \tag{2}$$

where R is reflectance and the subscripts indicate wavelengths. Here, we used the averages of 770-800 nm for NIR,
630-670 nm for red reflectance, and 460-475 nm for blue bands reflectance and normalized to reduce noise.
We further calculated the near-infrared vegetation index NIRv as:

$$NIRv = NDVI \times R_{770\text{-}800} \tag{3}$$

where R770-800 is the NIR reflectance (Badgley et al., 2017). The fluorescence correction vegetation index (FCVI)
was calculated from spectral data by subtracting the reflectance in the visible range (R400-700) from the NIR
reflectance (Yang et al., 2020) as follows

$$FCVI = R_{770\text{-}800} - R_{400\text{-}700} \tag{4}.$$

The near-infrared radiance of vegetation (NIRvrad) was calculated similarly to the NIRv, except NDVI was multiplied
by the radiance, rather than reflectance, from the NIR region (Rad770-800) (Wu et al., 2020) as follows:

$$NIRvrad\ =\ NDVI\ \times Rad_{770\text{-}800} \hspace{4cm} (5).$$

## 2.4 Data Analysis

A workflow summarizing data analyses is provided in Fig.1. We examined mean values across the canopy over the course of one day by creating a diurnal time series of scatterplots of the tower-based PAR data, tower-based GPP data, and means of all spectral vegetation indicators, on Jan 31, 2019, and ran comparisons using Pearson's correlation coefficients to examine correlations. Results are provided in Section 3.1 and Fig. 2. At fine spatial scales, i.e. pixel sizes of ~15 cm, we created density plots, calculated the coefficient of variation (CV), and calculated the means of all vegetation indicators (NDVI, EVI, NIRv, FCVI, NIRvrad) for each flight to compare spatial and temporal variability. Results are provided in Section 3.2 and Fig. 3. To determine which spatial scales dominate the variability of each vegetation quantity, we ran power spectrum wavelet analysis using code created in the Matlab programming language (Mathworks, Natick, Massachusetts). For each vegetation quantity and each flight, and for the lidar elevation model representing canopy height, we computed the Morlet wavelet power spectrum of individual transects (Torrence and Compo, 1998). All power spectra from the wavelet analysis were normalized to unit variance. An ensemble power spectrum for each vegetation indicator was created by averaging across all the transects of each flight and then across flights. We then compared the power spectra for each vegetation indicator and lidar data to compare the spatial scales at which the quantities captured variability as well as the spatial scale at which the lidar-based elevation model captured variability. Results are provided in Section 3.3 and Fig. 4. For illustration purposes, Fig. S3 is an example of two synthetic signals generated with fractal Brownian motion algorithm and different level of noise-to-signal ratio (Signal A and B, respectively, Fig. A1) and the corresponding power spectra which decay differently at smaller spatial scales (Power Spectra, Fig. A1). Initial UAS data processing was carried out in Interactive Data Language (IDL) and Environment for Visualizing Images (ENVI) (Harris Geospatial, Boulder, CO). Other analyses, including graphical illustrations, were carried out using the R open source environment with libraries dplyr, ggplot, and tidyverse (R Development Core Team, 2010; Wickham et al., 2018; Wickham, 2017, 2016) and Matlab R2019a (Mathworks, Natick, Massachusetts).

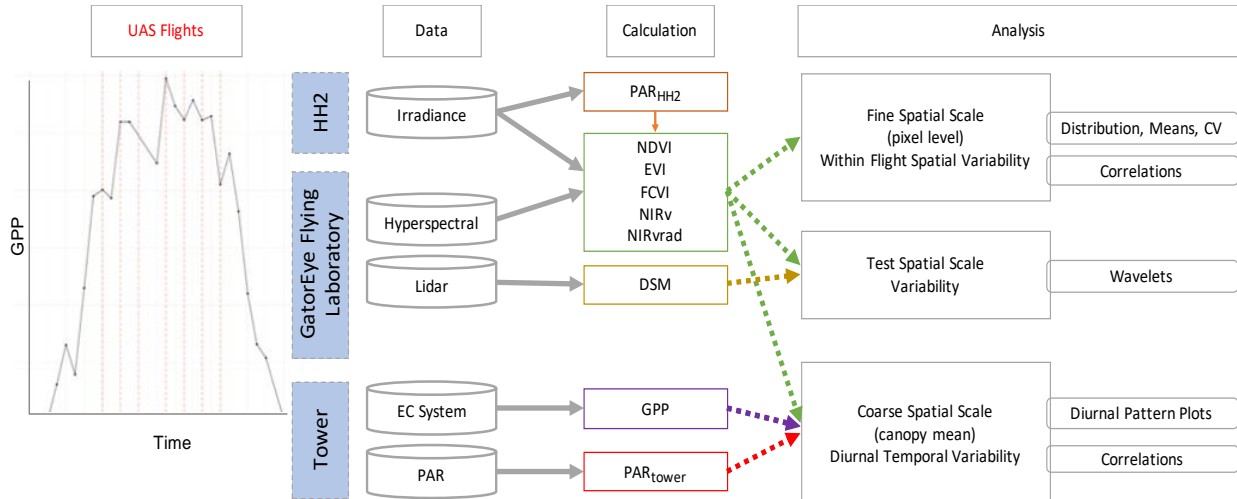

**Figure 1. Summary of methods. Diagram representing discrete flight times for UAS and near-continuous EC-estimated GPP (far left). Platforms and instrumentation (blue) consisted of the Analytical Spectral Devices (ASD) Handheld Spectroradiometer Pro 2 (HH2), the GatorEye Flying Laboratory, and the tower at Barro Colorado Island (BCI). Data collected included irradiance, hyperspectral, Lidar, Eddy Covariance System (EC), and Photosynthetically Active Radiation (PAR). Calculations made were PAR with the HH2 (PARHH2), the Normalized Difference Vegetation Index (NDVI), Enhanced Vegetation Index (EVI), Fluorescence Correction Vegetation Index (FCVI), the Near Infrared Vegetation Index (NIRv), the Near Infrared Radiance of Vegetation (NIRvrad), the Digital Surface Model (DSM), Gross Primary Productivity (GPP) and PAR from the PAR Sensor on the tower (PARtower). An overview of the data analysis at each scale is provided in the right of the diagram.**

## 3   Results and discussion

### 3.1 Diurnal trend in spectral vegetation indicators, PAR, and GPP

The degree to which remote sensing vegetation indicators represent changes in GPP depend largely on canopy structure-dependent light absorption and scattering processes, that is, exploiting relationships between a remote sensing vegetation quantity, PAR or APAR, and GPP. Fig. 2 shows GPP, PAR, and the mean value of each vegetation quantity at each flight time over the course of January 31, the day on which we had overlapping data between the UAS and eddy covariance system (Fig. 2a-d). Additionally, Pearson correlation coefficients among mean NIRv, FCVI, NIRvrad, EVI, and NDVI for each flight time and the GPP and PAR values at the flight times are shown in Fig. 2d. NIRv is significantly and strongly positively correlated to both FCVI (r=0.9, p<0.001) and EVI (r=0.9, p<0.01). NIRvrad is the only vegetation quantity with a significant correlation to PAR and GPP, with a strong positive relationship (0.9 and 0.81, respectively, p-values <0.05; Fig. 2d). Mean NIRvrad values also have the greatest relative diurnal change among the vegetation indicators (Fig. 2c and d). These results demonstrate that a shared correlation of NIRvrad and GPP to PAR results in mean NIRvrad tracking diurnal changes in GPP to a greater degree than NIRv, FCVI, NDVI or EVI, because NIRvrad takes incoming radiation into account whereas the other vegetation indicators do not. The ability of NIRvrad to track APAR is notable alone. However, our evidence supports the proposed use of NIRvrad as a proxy for changes in GPP on short timescales – albeit based on only one day of data. NIRvrad is a more practical proxy of GPP than SIF in the sense that a separate instrument to measure PAR is not needed (Wu et al., 2020; Zeng et al., 2019). Given that the relationship between NIRvrad and GPP depends on PAR, it is unclear if the

230  association between NIRvrad and GPP would weaken during the wet season when low light or diffuse light conditions
231  are more common (Berry and Goldsmith, 2020).

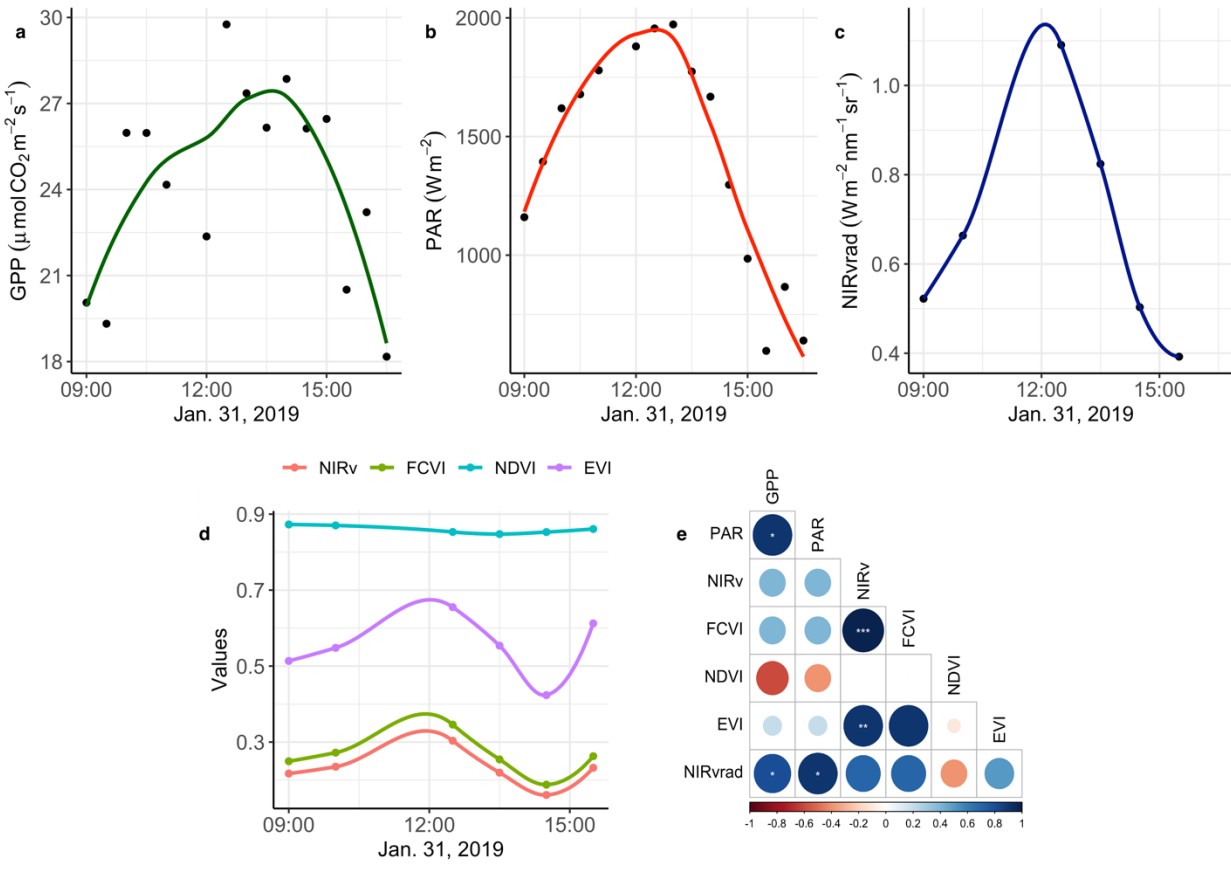

232

**Fig. 2. Diurnal time series smoothed with a LOESS filter of a) GPP b) PAR c) NIRvrad d) NIRv, FCVI, NDVI, and EVI e)**
**comparisons of quantities using Pearson correlations color indicates strength of relationship, * = p-value<0.05, ** = p-value**
**<0.01, *** = p-value <0.001.**

### 3.2 Tropical forest canopy variation

Spatial distributions and the coefficient of variation (CV) of all pixels of NIRv, FCVI, and NIRvrad are
generally similar to one another and show considerable variation spatially across the canopy and temporally over the
course of a day and across days (Fig. 3a-c, Table A2). NIRv, FCVI, and NIRvrad distributions are distinct from EVI
and NDVI (Fig. 3a-e, Table A2, and Table A2). NIRv, FCVI, and NIRvrad have the highest CV at each flight time
(between 39.78% and 91.54%, Table A1), followed by EVI (between 20.24% and 37.24%, Table A2) and NDVI
varied the least at any flight time (between 9.83% and 12.82%, Table A2). For some indices, mean values across the
canopy fail to capture extreme high (NIRv, NIRvrad, and FCVI) or low values (NDVI) during morning and afternoon
hours. This pattern suggests "hot" and "cool" spots of activity related to heterogeneity in forest structure and low sun
angles. In previous studies, the directional effects on NIRv have been examined on coarse spatial scales (i.e. satellites)
and have been proposed as a means of improving understanding of NIRv agreement to GPP (Hao et al., 2021; Dechant
et al., 2020; Baldocchi et al., 2020; Zhang et al., 2020). Our results demonstrate that NIRv, FCVI, and NIRvrad capture
fine-grained heterogeneity of this tropical forest canopy, which was obscured by EVI and NDVI (Fig. 3a-e). NIRv

and NIRvrad use NDVI, thus, by definition, NIR is the largest contributing factor to the heterogeneity captured (Fig. 3a, c, and e). While NIRv and NIRvrad distributions are generally similar, they diverge in the afternoons when PAR declines, which likely is why NIRvrad is better correlated with GPP. EVI variability was higher than NDVI variability, but lower than that of NIRv, FCVI, and NIRvrad, indicating that EVI has a different level of sensitivity to viewing geometry and canopy components (potentially understory), light absorption and scattering regime of the canopy than the other indices (Table A1and Table A2). We also show empirically that NIRv and FCVI are virtually the same in a dense tropical forest presumably due to both indices similarly representing the radiation regime of the tropical forest canopy, i.e. light capture and scattering, in conditions with little background soil, supporting the predictions of earlier studies (Dechant et al., 2020; Zeng et al., 2019; Yang et al., 2018b; Wu et al., 2020).

Midday distributions of NIRv, FCVI, and NIRvrad on Jan. 30 at 1200 and 1330 and Jan. 31 at 1230 are less skewed than at other times of the day whereas morning and afternoon distributions are skewed toward lower values, except for Jan. 31 at 1530 (Fig. 3a-c). On both days, when mean values peak at midday, the variation for all vegetation indicators is lowest (Jan 30, 1200 CV between 47.6 and 49.2 and Jan 31, 1230 CV between 45.6 and 47.2) (Fig. 3, Table A1). The highest variability occurred in the afternoon on both days (Jan 30, 1630 CV between 91.3% and 91.5 and Jan 31, 1430 CV between 83.3% and 83.8% for all quantities) (Fig. 3, Table A2). At midday, NIRv, FCVI, and NIRvrad variability was low and means were high, indicating that viewing and sun geometry drive the higher and lower values during morning and afternoon. This effect is greater in the afternoon than the morning (Fig. 3, Table A2). However, a different pattern is apparent on Jan. 31 during the 1530 flight time when mean values increased from the 1430 flight time means and the CV values were the lowest of any flight observations in the study and this influence appears to be greatest on EVI. It is possible that this was due to another type of effect on illumination geometry, such as wind influencing the UAS, diffuse radiation effects, or hotspot effects.

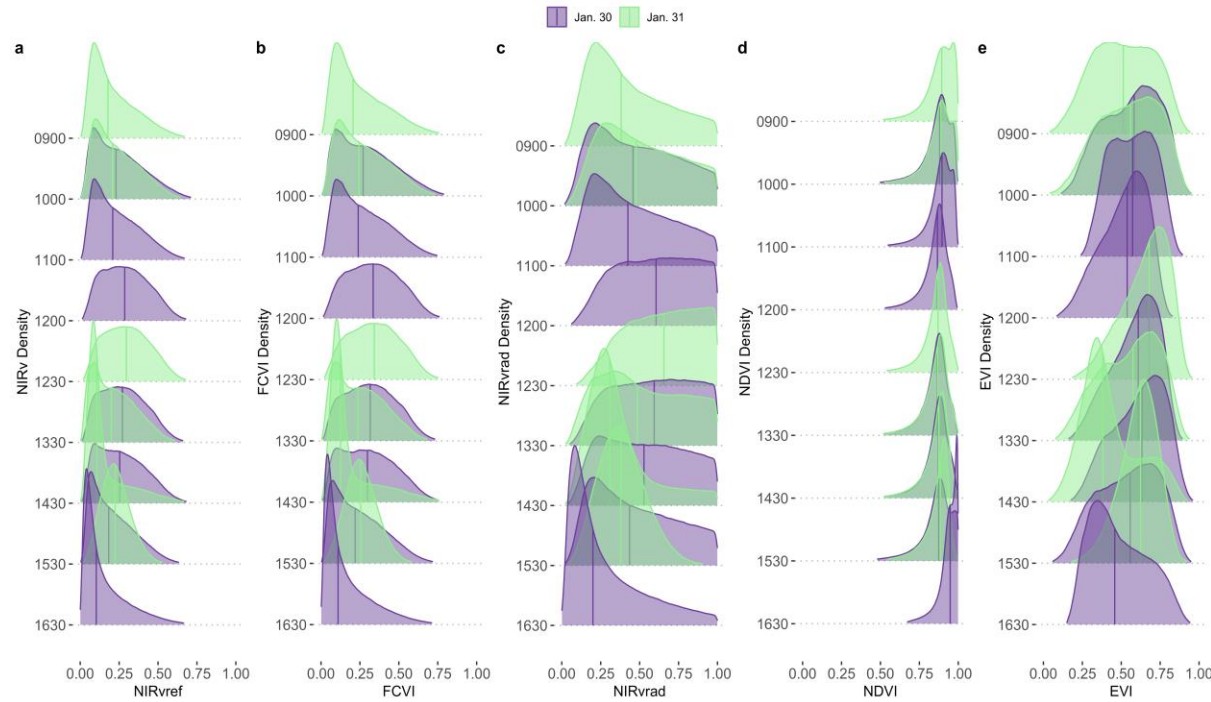

**Fig. 3. NIRv (a), FCVI (b), and NIRvrad (c) density plots for each flight time on January 30 and January 31, 2019. Colours of distributions indicate day.**

**3.3 Power Spectrum Analysis**

Power spectrum analysis was used to identify the dominant spatial scales driving variability across the canopy (Fig. 4). In Fig. 4, the area beneath the curve is proportional to the variance because it is the spectrum divided by the corresponding scale and then plotted as a function of the log of the scale (example signals and power spectra provided Fig. A1). Similar to their spatial distributions (Fig. 3), NIRvrad and FCVI are indistinguishable in their dominant scales of spatial variability (Fig. 3) (Dechant et al., 2020; Zeng et al., 2019). Power spectrum analysis shows a distinct peak around 50 m spatial scale for NIRv, NIRvrad, FCVI, and EVI, whereas NDVI peaks at approximately 90 m. The largest tree crown sizes on BCI are on the order of 20-30 m in diameter and the most common crown sizes are between 4-10 m (Fig. A2). Thus, the spatial variability of the vegetation indicators is strongly influenced by larger forest structures, such as forest gaps and tree clusters, rather than individual tree crowns.

This larger scale of variability is also confirmed by the power spectrum of the lidar-derived canopy surface model, which displays a peak at 70 m scale, indicating that larger than tree crown scales produce the most variability in canopy height. In other words, UAS-based lidar data also show that canopy heights within a 70 m spatial scale create strong spatial features on the landscape. Vegetation indicators and the lidar canopy surface model appear less effective at capturing smaller scale differences within a canopy (leaves or leaf clumps) or among the most frequent tree crown sizes on BCI (4-10 m sunlit tree crown sizes determined by stereophotos; Fig. A2). However, the peaks in the vegetation indicators are broader than the peak in the lidar data, showing that smaller features of the canopy are still contributing to the total spatial signal in the power spectra. These results suggest that satellite data with a spatial resolution greater than ~50 m may miss important variation in diverse tropical forest canopies. NDVI displays a

different shape with a slower decay at small scales, indicating less distinguishable spatial structures from the canopy,
and a peak shifted to the larger scales (Fig. 4), i.e. NDVI does not distinguish smaller spatial structures. At much larger
scales (>100-200 m), the vegetation indicators decline smoothly, while NDVI and especially lidar show an increase
in variance probably associated with topographic heterogeneity.

296        One reason why vegetation indicators and LiDAR captured variability at spatial scales larger than the most

common tree crown sizes on BCI is that canopy heights tend to be more uniform on BCI compared to other tropical
forests, possibly due to wind (Bohlman and O'Brien, 2006). For example, Dipterocarpus dominated South-East Asian
forests have emergent trees, unlike BCI, which can reach up to 60 m in height. Additionally, tree crowns on BCI tend
to be more flat-topped than conical or rounded, and trees can be found clumped in similar heights, which could explain
why the most often detected unit is larger than the mean of a single crown. On the other end of the spectrum, forest
gaps can be larger than a single crown because treefall often affects neighbouring trees.

303        Vegetation indicators and the Lidar-derived surface model represent the spectral and structural properties most

broadly of the upper canopy, and thus it is conceivable that they display similar spatial variability. However, NIRv,
FCVI, NIRvrad, and EVI discriminated details at a different spatial scale from NDVI and LiDAR. These results
parallel the variability detected in their distributions (Fig. 3 and Table A1), where NDVI patterns were distinct from
the other vegetation indicators. Taken together, these results show that NIRv, FCVI, and NIRvrad have a smoother
spatial pattern and peak at finer scales than NDVI, which is known to saturate at high green biomass (Zhu and Liu,
2015; Huete et al., 2002), whereas NIRv, FCVI, and NIRvrad should better correlate with aspects of photosynthetic
capacity. Thus, these emerging indicators should measure finer resolution spatial heterogeneity and should be more
adept at monitoring changes in structure and function of the canopy than NDVI. Additionally, the emerging indicators
can potentially disaggregate the physiological and structural component of SIF when SIF measurements are available
since changes in structure of the forest coincide with changes in GPP (Wang et al., 2020; Wu et al., 2020; Yang et al.,
2020; Dechant et al., 2020). Emerging indicators' heightened ability to differentiate the fine-scale spatial variability
in the canopy is likely due to the influence of high upwelling of NIR from the canopy and understory, particularly in
the dry season, which tend to blur the signal of the upper canopy for NDVI. Notably, EVI and NDVI, two common
indicators of vegetation greenness, show differences in their power spectrum, in particular the slope of the curve for
scales less than 20 m. EVI was designed to better capture vegetation changes by exploiting variability in the reflectance
in the blue range, especially effective in dense green canopies. This may help explain the scale of variability in this
canopy where variation in the blue may be expected to manifest, especially because deciduous crowns, which have
high reflectance in blue wavelengths compared to fully leaved crowns, are present on BCI (Bohlman, 2008).

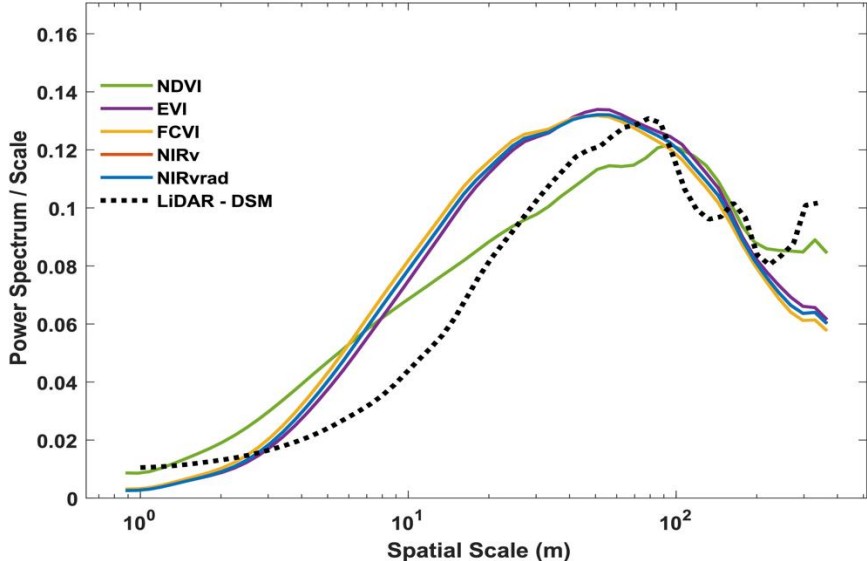


**Fig. 4. Ensemble wavelet power spectra for all the quantities used in this study and a LiDAR-derived digital surface model (DSM). Note that FCVI and NIRv are similar, thus the NIRv curve is obscured by the FCVI. Ensembles were created by averaging the spectrum of individual transects, then averaging across flights. Note that in this representation, the spectrum divided by the corresponding scale as a function of the log of the scale, the area beneath the curve is proportional to the variance.**

## 4    Conclusions

We examined NIRv, FCVI, and NIRvrad, emerging vegetation indicators related to fPAR of a semi-deciduous tropical forest canopy using UAS-based hyperspectral data. Our findings demonstrate that NIRvrad has greater potential to track GPP over the course of a day than the non-radiance-based indices as evidenced by a shared correlation among NIRvrad, PAR, and GPP. Thus, NIRvrad is a potential proxy for tracking GPP on short timescales without the need for separate measurements of incoming irradiance. Also, NIRv, FCVI, and NIRvrad at high spatial resolution (~15cm) unveil greater spatial and diurnal variability of BCI's tropical forest canopy versus EVI or NDVI, which may pave the way to improve our understanding of the relationship between GPP and remote sensing observations. For instance, by benchmarking changes of vegetation function and structure that underlie a GPP measurement representing the whole EC footprint, fine scale NIRv, FCVI, or NIRvrad measurements may reveal highly differential behaviors of tropical species diurnally to seasonally. The dominant scale driving spatial variability of spectral measurements and lidar data are larger forest structures occurring on BCI, such as groups of similar trees or forest gaps. Yet, smaller, broader peaks in the power spectra of NIRv, FCVI, NIRvrad, and EVI indicate these four indices incorporate smaller scale information compared to NDVI. Taken together, the demonstrated potential to track GPP, measure spatial heterogeneity and variability, and capture forest structural characteristics of BCI open greater possibilities to examine structure and function within and across this tropical forest.

Because remote sensing advancements are making it possible to capture physiological responses of vegetation, the importance of improved techniques to examine the radiation regime, for instance estimating fPAR or APAR, can

be overlooked. However, recent studies have highlighted the importance and difficulties of measuring fPAR and
APAR, the strong dependence of measurements on illumination and viewing geometry, as well as the need for
increased understanding of structure-related radiation regime information more generally e.g. (Hao et al., 2021;
Dechant et al., 2020; Baldocchi et al., 2020; Rocha et al., 2021; Zhang et al., 2020). For NIRv, FCVI, and NIRvrad,
inclusion of the NIR spectral region makes the emerging indices more sensitive to incoming, absorbed, and scattered
radiation, which can be influenced by illumination and viewing geometry, changes in canopy leaf angles or associated
structure changes. In the case of NIRvrad, which was most strongly associated with GPP, changes in light regime and
associated photosynthetic capacity can even be captured diurnally. Furthermore, NIRv, FCVI, and NIRvrad
measurements, especially at high spatial and temporal resolution can help inform our understanding of one another,
traditional reflectance-based indices, and other measurements such as SIF. This study highlights the importance of
understanding the incoming solar radiation, absorbed and scattered radiation, and illumination and viewing geometry
of any remote sensing data, but it also encourages exploiting RS observations to improve our ability to measure
structure-related light capture and scattering patterns. It is in this role, we show these measurements should be further
investigated as valuable tools to improve our understanding of complex tropical forest canopies and potentially as an
improved estimate of fPAR, APAR, or GPP. While this study focuses on BCI, these techniques could be applied more
broadly for the purposes of defining the dominant scale of spatial variability, tracking structural changes, monitoring
coincident changes in GPP or light regime, or as inputs to vegetation models of tropical forest structure and function.
**5    Appendix A**

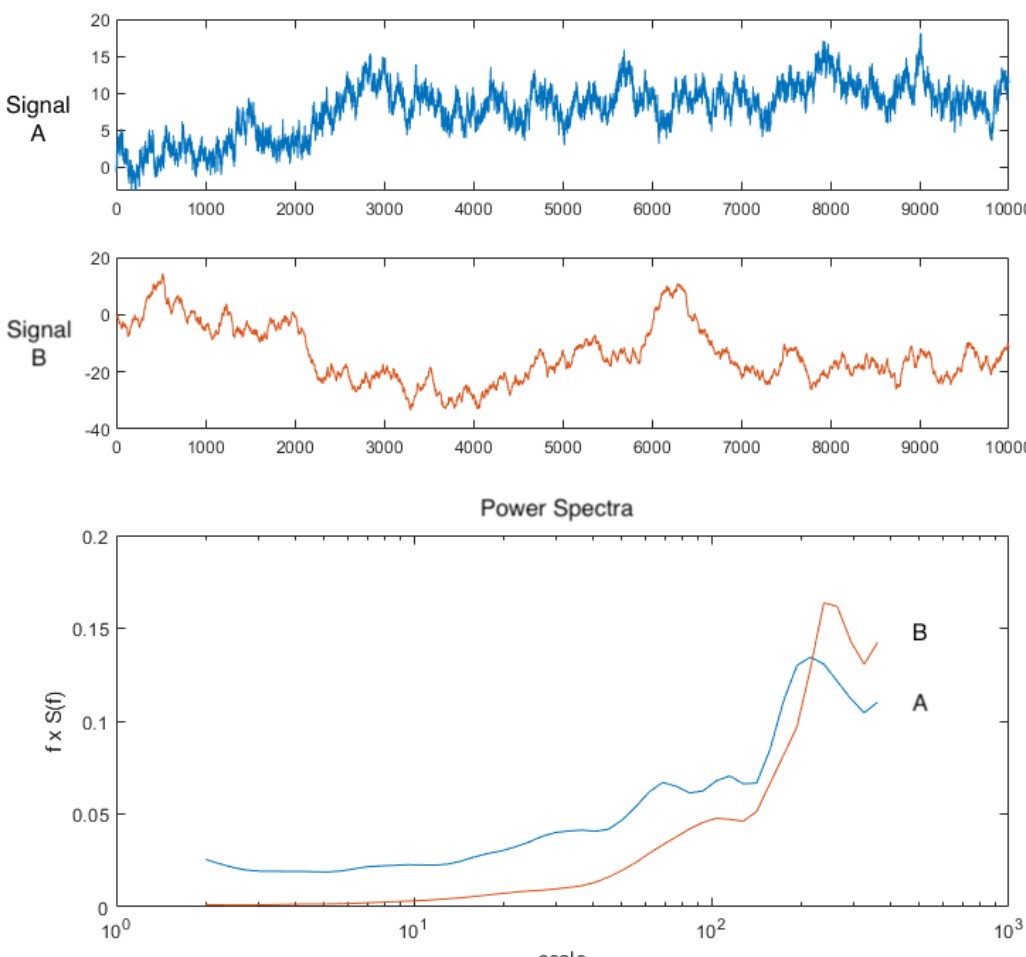


**Figure A1. Sample signals with relatively higher noise (Signal A) and lower noise (Signal B) and their corresponding Power**
**Spectra ensemble plotted as normalized on log scale. Note the representation of the variance by area under the curve is**
**preserved by multiplying the Power (S(f)) by the frequency (f). In this way the area beneath the curve is still proportional**
**to the variance.**

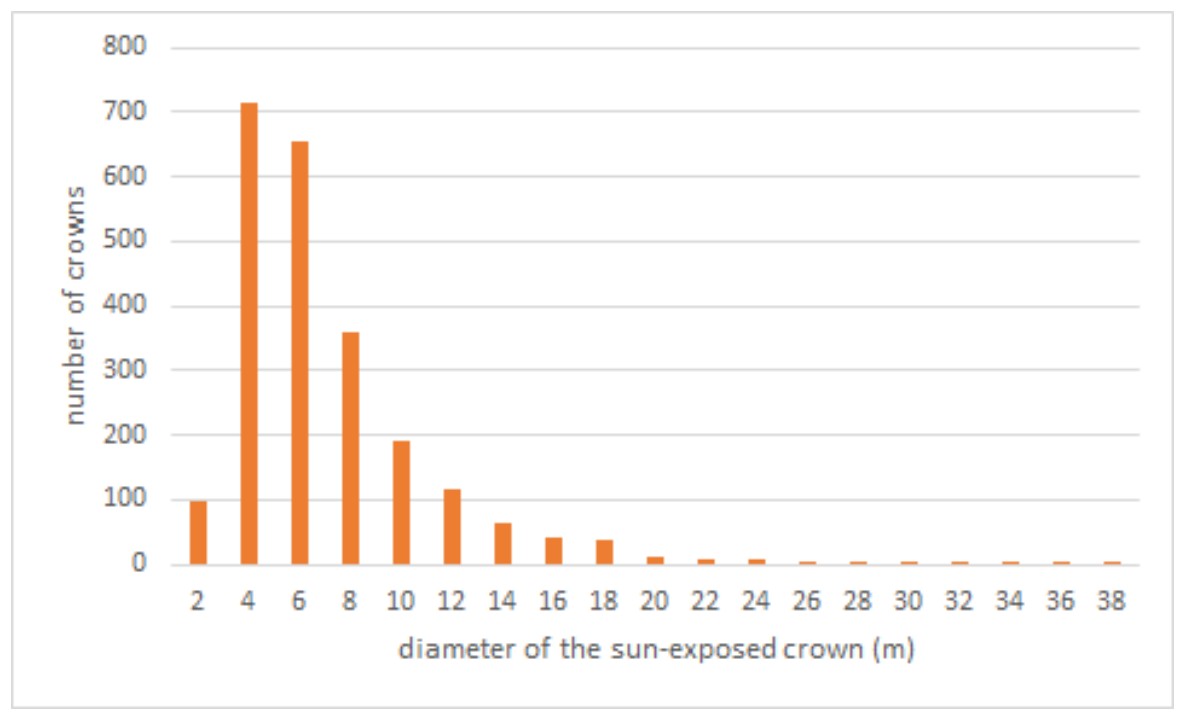


**Figure A2. Distribution of tree crown sizes on BCI in a sample ~10 ha plot taken from digitized high spatial resolution**
**stereo photos that were linked to stems in the field (Bohlman and Pacala 2012). This ~10 ha plot does not coincide with the**
**~10 ha area sampled by the UAS near the eddy covariance tower in this study.**

**Table A1. Mean, standard deviation (Sdev) and coefficient of variation (CV) of NIRv, NIRvrad, and FCVI measurements**
**for the study.**

| Flight Time | Mean NIRv | SDev NIRv | CV NIRv (%) | Mean NIRvrad | SDev NIRvrad | CV NIRvrad (%) | Mean FCVI | SDev FCVI | CV FCVI (%) |
|---|---|---|---|---|---|---|---|---|---|
| Jan30_1000 | 0.26 | 0.16 | 61.36 | 0.60 | 0.36 | 60.54 | 0.29 | 0.18 | 59.69 |
| Jan30_1100 | 0.24 | 0.15 | 61.48 | 0.54 | 0.33 | 60.56 | 0.27 | 0.16 | 60.89 |
| Jan30_1200 | 0.29 | 0.15 | 49.20 | 0.82 | 0.39 | 47.59 | 0.34 | 0.16 | 47.88 |
| Jan30_1330 | 0.28 | 0.14 | 50.46 | 0.81 | 0.40 | 49.24 | 0.32 | 0.16 | 49.16 |
| Jan30_1430 | 0.27 | 0.15 | 55.46 | 0.70 | 0.38 | 54.38 | 0.31 | 0.17 | 54.22 |
| Jan30_1530 | 0.21 | 0.14 | 65.10 | 0.63 | 0.41 | 64.71 | 0.25 | 0.16 | 64.01 |
| Jan30_1630 | 0.16 | 0.14 | 91.54 | 0.32 | 0.30 | 91.54 | 0.17 | 0.15 | 91.39 |
| Jan31_0900 | 0.22 | 0.14 | 66.31 | 0.52 | 0.34 | 65.25 | 0.25 | 0.16 | 66.01 |
| Jan31_1000 | 0.24 | 0.14 | 59.43 | 0.66 | 0.39 | 58.29 | 0.27 | 0.16 | 59.04 |
| Jan31_1230 | 0.30 | 0.14 | 47.17 | 1.09 | 0.50 | 45.63 | 0.35 | 0.16 | 45.91 |
| Jan31_1330 | 0.22 | 0.14 | 61.91 | 0.82 | 0.51 | 61.47 | 0.25 | 0.15 | 60.53 |
| Jan31_1430 | 0.16 | 0.14 | 85.32 | 0.50 | 0.42 | 83.81 | 0.19 | 0.16 | 83.83 |
| Jan31_1530 | 0.86 | 0.08 | 9.83 | 0.61 | 0.12 | 20.24 | 0.53 | 0.04 | 8.15 |


**Table A2. Mean, standard deviation (Sdev) and coefficient of variation (CV) of NDVI and EVI measurements for the study.**

| Flight Time | Mean NDVI | SDev NDVI | CV NDVI (%) | Mean EVI | SDev EVI | CV EVI (%) |
|---|---|---|---|---|---|---|
| Jan30_1000 | 0.86 | 0.10 | 11.64 | 0.57 | 0.18 | 31.54 |
| Jan30_1100 | 0.88 | 0.09 | 10.15 | 0.57 | 0.14 | 24.40 |
| Jan30_1200 | 0.85 | 0.09 | 10.38 | 0.52 | 0.15 | 28.48 |
| Jan30_1330 | 0.85 | 0.09 | 10.60 | 0.59 | 0.15 | 25.24 |
| Jan30_1430 | 0.85 | 0.09 | 10.35 | 0.61 | 0.16 | 26.84 |
| Jan30_1530 | 0.85 | 0.11 | 12.52 | 0.54 | 0.19 | 35.21 |
| Jan30_1630 | 0.93 | 0.06 | 6.69 | 0.49 | 0.18 | 36.90 |
| Jan31_0900 | 0.87 | 0.10 | 11.54 | 0.51 | 0.19 | 37.24 |
| Jan31_1000 | 0.87 | 0.10 | 11.08 | 0.55 | 0.19 | 34.66 |
| Jan31_1230 | 0.85 | 0.08 | 9.82 | 0.66 | 0.15 | 22.72 |
| Jan31_1330 | 0.85 | 0.09 | 10.70 | 0.55 | 0.19 | 33.80 |
| Jan31_1430 | 0.85 | 0.09 | 10.58 | 0.42 | 0.18 | 43.07 |
| Jan31_1530 | 0.86 | 0.08 | 9.83 | 0.61 | 0.12 | 20.24 |



*Code availability*
*Data availability*
GatorEye data related to this project can be downloaded from www.gatoreye.org. Code and other material
with links provided upon request.

*Author contributions*
T.M. designed the study with the help of S.P. and S.A.B. while a Provost's Postdoctoral Fellow at Florida State
University. T.M. performed field work, data collection, processing, and initial manuscript submission a Provost's
Postdoctoral Fellow at Florida State University. M.D. and T.M. outfitted the tower and collected tower-based data,
T.M. and E.N.B. collected the UAS data. E.N.B., A.M.A.Z., and T.M. pre-processed the hyperspectral and lidar data.
T.M. and M.D. further processed UAV, lidar, and GPP data and ran data analysis. M.D., S.P., S.A.B., C.S., contributed
with the methodological framework, data processing analysis and write up T.M., M.D., S.P., S.A.B., C.S., E.N.B., and
A.M.A.Z. contributed to the interpretation, quality control and revisions of the manuscript. All authors read and
approved the final version of the manuscript.
*Competing interests*
The authors declare no conflict of interest.
*Acknowledgments*
This project execution was carried out while T.M., the primary author, was a Provost's Postdoctoral Fellow
in the Department of Geography at Florida State University under the advisement of S.P.. T.M. wishes to extend the
sincerest thanks to S.P. for support and guidance, as well as to the FSU Department of Geography, FSU Provost's
Postdoctoral Fellows program, and to co-authors who served as mentors. Support for this project, including portions
of field logistic and data collection costs and materials, and support for T.M., was provided by the Provost's
Postdoctoral Fellows Program at Florida State University. E.N.B. was supported through the School of Forest,
Fisheries and Geomatics Sciences and expenses and data collection paid for by T.M.'s Provost's Postdoctoral Fellows
Program at Florida State University, A.M.A.Z through the Center for Latin American Studies, and hardware, software,
and system costs associated with the GatorEye and data collection were provided through the McIntire Stennis
Program of the USDA and the School of Forest, Fisheries and Geomatics Sciences. M.D. was supported by the Carbon
Mitigation Initiative at Princeton University. The authors wish to thank the vast support of the collaborators, staff, and
researchers at the Smithsonian Tropical Research Institute and, specifically at Barro Colorado Island, without which
this research would not be possible. Among other contributors to the work, we also extend special thanks to Alfonso
Zambrano, Carli Merrick, Riley Fortier, and Pete Kerby-Miller for field work assistance, and Dr. S. Joseph Wright
and Dr. Helene Muller-Landau for support on site as well.

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

Index (MTVDI) Captures Canopy Seasonality across Amazonian Tropical Evergreen Forests, Remote Sensing, 13,
10.3390/rs13030339, 2021.
Liu, X., Liu, L., Zhang, S., and Zhou, X.: New Spectral Fitting Method for Full-Spectrum Solar-Induced Chlorophyll
Fluorescence Retrieval Based on Principal Components Analysis, Remote Sensing, 7, 10626-10645,
10.3390/rs70810626, 2015.
Logan, B. A., Adams, W. W., and Demmig-Adams, B.: Viewpoint:Avoiding common pitfalls of chlorophyll
fluorescence analysis under field conditions, Functional Plant Biology, 34, 853, 10.1071/fp07113, 2007.
Magney, T. S., Frankenberg, C., Fisher, J. B., Sun, Y., North, G. B., Davis, T. S., Kornfeld, A., and Siebke, K.:
Connecting active to passive fluorescence with photosynthesis: a method for evaluating remote sensing measurements
of Chl fluorescence, The New phytologist, 1594-1608, 10.1111/nph.14662, 2017.
Malenovsky, Z., Mishra, K. B., Zemek, F., Rascher, U., and Nedbal, L.: Scientific and technical challenges in remote
sensing of plant canopy reflectance and fluorescence, Journal of experimental botany, 60, 2987-3004,
10.1093/jxb/erp156, 2009.
Malhi, Y.: The productivity, metabolism and carbon cycle of tropical forest vegetation, Journal of Ecology, 100, 65-
75, 10.1111/j.1365-2745.2011.01916.x, 2012.
Medlyn, B. E.: Physiological basis of the light use efficiency model, Tree Physiology, 18, 167–176,
https://doi.org/10.1093/treephys/18.3.167, 1998.
Meroni, M., Rossini, M., Guanter, L., Alonso, L., Rascher, U., Colombo, R., and Moreno, J.: Remote sensing of solar-
induced chlorophyll fluorescence: Review of methods and applications, Remote Sensing of Environment, 113, 2037-
2051, 10.1016/j.rse.2009.05.003, 2009.
Merrick, Pau, Jorge, Bennartz, and Silva: Spatiotemporal Patterns and Phenology of Tropical Vegetation Solar-
Induced Chlorophyll Fluorescence across Brazilian Biomes Using Satellite Observations, Remote Sensing, 11,
10.3390/rs11151746, 2019.
Merrick, T., Jorge, M. L. S. P., Silva, T. S. F., Pau, S., Rausch, J., Broadbent, E. N., and Bennartz, R.: Characterization
of chlorophyll fluorescence, absorbed photosynthetically active radiation, and reflectance-based vegetation index
spectroradiometer measurements, International Journal of Remote Sensing, 41, 6755-6782,
594 10.1080/01431161.2020.1750731, 2020.
Mitchard, E. T. A.: The tropical forest carbon cycle and climate change, Nature, 559, 527-534, 10.1038/s41586-018-
596 0300-2, 2018.
Monteith, J.L., Climate and the efficiency of crop production in Britain, Phil. Trans. R. Soc. Land., 281, 277-294,
598 1977.
Morton, D. C., Rubio, J., Cook, B. D., Gastellu-Etchegorry, J. P., Longo, M., Choi, H., Hunter, M. O., and Keller, M.:
Amazon forest structure generates diurnal and seasonal variability in light utilization, Biogeosciences Discussions,
12, 19043-19072, 10.5194/bgd-12-19043-2015, 2015.
Morton, D. C., Nagol, J., Carabajal, C. C., Rosette, J., Palace, M., Cook, B. D., Vermote, E. F., Harding, D. J., and
North, P. R.: Amazon forests maintain consistent canopy structure and greenness during the dry season, Nature, 506,
221-224, 10.1038/nature13006, 2014.

Moya, I., Camenen, L., Evain, S., Goulas, Y., Cerovic, Z. G., Latouche, G., Flexas, J., and Ounis, A.: A new instrument for passive remote sensing1. Measurements of sunlight-induced chlorophyll fluorescence, Remote Sensing of Environment, 91, 186-197, 10.1016/j.rse.2004.02.012, 2004.

Plascyk, J. A.: The MK II Fraunhofer Line Discriminator (FLD -II) for Airborne and Orbital Remote Sensing of Solar-Stimulated Luminescense, Optical Engineering, 14, 8, 1975.

Porcar-Castell, A., Tyystjarvi, E., Atherton, J., van der Tol, C., Flexas, J., Pfundel, E. E., Moreno, J., Frankenberg, C., and Berry, J. A.: Linking chlorophyll a fluorescence to photosynthesis for remote sensing applications: mechanisms and challenges, Journal of experimental botany, 65, 4065-4095, 10.1093/jxb/eru191, 2014.

R Development Core Team: R: A language and environment for statistical computing, R Foundation for Statistical Computing [code], 2010.

Rocha, A. V., Appel, R., Bret-Harte, M. S., Euskirchen, E. S., Salmon, V., and Shaver, G.: Solar position confounds the relationship between ecosystem function and vegetation indices derived from solar and photosynthetically active radiation fluxes, Agricultural and Forest Meteorology, 298-299, 10.1016/j.agrformet.2020.108291, 2021.

Rong Li, F. Z.: Accuracy assessment on reconstruction algorithms of solar-induced Fluorescence Spectrum, Geoscience and Remote Sensing Symposium (IGARSS) IEEE International, 1727-1730,

Rossini, M., Alonso, L., Cogliati, S., Damm, A., Guanter, L., Julitta, T., Meroni, M., Moreno, J., Panigada, C., Pinto, F., Rascher, U., Schickling, A., Schüttemeyer, D., Zemek, F., and Colombo, R.: Measuring sun-induced chlorophyll fluorescence: An evaluation and synthesis of existing field data, 5th International workshop on remote sensing of vegetation fluorescence, Paris, France, 1-5,

Rouse Jr, J. W., Haas, R. H., Schell, J. A., and Deering, D. W.: Paper A 20, hird Earth Resources Technology Satellite-1 Symposium: The Proceedings of a Symposium Goddard Space Flight Center at Washington, DC 309,

Running, S. W., Nemani, R. R., Heinsch, F. A., Zhao, M., Reeves, M., and Hashimoto, H.: A Continuous Satellite-Derived Measure of Global Terrestrial Primary Production, BioScience, 54, 547-551, 2004.

Ryu, Y., Jiang, C., Kobayashi, H., and Detto, M.: MODIS-derived global land products of shortwave radiation and diffuse and total photosynthetically active radiation at 5 km resolution from 2000, Remote Sensing of Environment, 204, 812-825, 10.1016/j.rse.2017.09.021, 2018.

Saatchi, S. S., Harris, N. L., Brown, S., Lefsky, M., A., E. T., Mitchare, W. S., Zutta, B. R., Buerman, W., Lewis, S. L., Hagen, S., Petrova, S., White, L., Silman, M., and Morel, A.: Benchmark map of forest carbon stocks in tropical regions across three continents, Proceedings of the National Academy of Sciences, 108, 9899-9905, 2010.

Saatchi, S. S., Harris, N. L., Brown, S., Lefsky, M., Mitchard, E. T., Salas, W., Zutta, B. R., Buermann, W., Lewis, S. L., Hagen, S., Petrova, S., White, L., Silman, M., and Morel, A.: Benchmark map of forest carbon stocks in tropical regions across three continents, Proceedings of the National Academy of Sciences of the United States of America, 108, 9899-9904, 10.1073/pnas.1019576108, 2011.

Samanta, A., Ganguly, S., and Myneni, R.: MODIS Enhanced Vegetation Index data do not show greening of Amazon forests during the 2005 drought, New Phytologist, 189, 4, 2010.

Schickling, A., Matveeva, M., Damm, A., Schween, J., Wahner, A., Graf, A., Crewell, S., and Rascher, U.: Combining Sun-Induced Chlorophyll Fluorescence and Photochemical Reflectance Index Improves Diurnal Modeling of Gross Primary Productivity, Remote Sensing, 8, 574, 10.3390/rs8070574, 2016.

Sims, D., Rahman, A., Cordova, V., Elmasri, B., Baldocchi, D., Bolstad, P., Flanagan, L., Goldstein, A., Hollinger, D., and Misson, L.: A new model of gross primary productivity for North American ecosystems based solely on the enhanced vegetation index and land surface temperature from MODIS, Remote Sensing of Environment, 112, 1633-1646, 10.1016/j.rse.2007.08.004, 2008.

Springer, K., Wang, R., and Gamon, J. A.: Parallel Seasonal Patterns of Photosynthesis, Fluorescence, and Reflectance Indices in Boreal Trees, Remote Sensing, 9, 1-18, 10.3390/rs9070691, 2017.

Sun, Y., Frankenberg, C., Wood, J. D., Schimel, D. S., Jung, M., Guanter, L., Drewry, D. T., Verma, M., Porcar-Castell, A., Griffis, T. J., Gu, L., Magney, T. S., Kohler, P., Evans, B., and Yuen, K.: OCO-2 advances photosynthesis observation from space via solar-induced chlorophyll fluorescence, Science, 358, eaam5747, 10.1126/science.aam5747, 2017.

Torrence, C. and Compo, G. P.: A Practical Guide to Wavelet Analysis, Bulletin of the American Meteorological Society, 79, 61-79, 1998.

Tucker, C., Red and photographic infrared linear combinations for vegetation monitoring, Remote Sensing of Environment, 8, 127-150, 1979.

Turner, D. P., Ritts, W. D., Cohen, W. B., Gower, S. T., Zhao, M., Running, S. W., Wofsy, S. C., Urbanski, S., Dunn, A. L., and Munger, J. W.: Scaling Gross Primary Production (GPP) over boreal and deciduous forest landscapes in support of MODIS GPP product validation, Remote Sensing of Environment, 88, 256-270, 10.1016/j.rse.2003.06.005, 2003.

Van Wittenberghe, S., Alonso, L., Verrelst, J., Moreno, J., and Samson, R.: Bidirectional sun-induced chlorophyll fluorescence emission is influenced by leaf structure and light scattering properties — A bottom-up approach, Remote Sensing of Environment, 158, 169-179, 10.1016/j.rse.2014.11.012, 2015.

Van Wittenberghe, S., Alonso, L., Verrelst, J., Hermans, I., Delegido, J., Veroustraete, F., Valcke, R., Moreno, J., and Samson, R.: Upward and downward solar-induced chlorophyll fluorescence yield indices of four tree species as indicators of traffic pollution in Valencia, Environmental pollution, 173, 29-37, 10.1016/j.envpol.2012.10.003, 2013.

Wang, C., Beringer, J., Hutley, L. B., Cleverly, J., Li, J., Liu, Q., and Sun, Y.: Phenology Dynamics of Dryland Ecosystems Along the North Australian Tropical Transect Revealed by Satellite Solar-Induced Chlorophyll Fluorescence, Geophysical Research Letters, 46, 5294-5302, 10.1029/2019gl082716, 2019.

Wang, S., Zhang, Y., Ju, W., Qiu, B., and Zhang, Z.: Tracking the seasonal and inter-annual variations of global gross primary production during last four decades using satellite near-infrared reflectance data, The Science of the total environment, 755, 142569, 10.1016/j.scitotenv.2020.142569, 2020.

Wickham, H.: ggplot2: Elegant Graphics for Data Analysis, Springer-Verlag [code], 2016.

Wickham, H.: tidyverse: Easily Install and Load the 'Tidyverse' (R package version 1.2.1) [code], 2017.

Wickham, H., François, R., Henry, L., and Müller, K.: dplyr: A Grammar of Data Manipulation (R package version 0.7.8) [code], 2018.

Wright, S. J.: The future of tropical forests, Ann N Y Acad Sci, 1195, 1-27, 10.1111/j.1749-6632.2010.05455.x, 2010.

Wu, G., Guan, K., Jiang, C., Peng, B., Kimm, H., Chen, M., Yang, X., Wang, S., Suyker, A. E., Bernacchi, C. J., Moore, C. E., Zeng, Y., Berry, J. A., and Cendrero-Mateo, M. P.: Radiance-based NIRv as a proxy for GPP of corn and soybean, Environmental Research Letters, 15, 10.1088/1748-9326/ab65cc, 2020.

Xu, L., Saatchi, S. S., Yang, Y., Myneni, R. B., Frankenberg, C., Chowdhury, D., and Bi, J.: Satellite observation of tropical forest seasonality: spatial patterns of carbon exchange in Amazonia, Environmental Research Letters, 10, 084005, 10.1088/1748-9326/10/8/084005, 2015.

Yang, H., Yang, X., Zhang, Y., Heskel, M. A., Lu, X., Munger, J. W., Sun, S., and Tang, J.: Chlorophyll fluorescence tracks seasonal variations of photosynthesis from leaf to canopy in a temperate forest, Glob Chang Biol, 23, 2874-2886, 10.1111/gcb.13590, 2017.

Yang, J., Tian, H., Pan, S., Chen, G., Zhang, B., and Dangal, S.: Amazon droughts and forest responses: Largely reduced forest photosynthesis but slightly increased canopy greenness during the extreme drought of 2015/2016, Glob Chang Biol, 1919-1934, 10.1111/gcb.14056, 2018a.

Yang, K., Ryu, Y., Dechant, B., Berry, J. A., Hwang, Y., Jiang, C., Kang, M., Kim, J., Kimm, H., Kornfeld, A., and Yang, X.: Sun-induced chlorophyll fluorescence is more strongly related to absorbed light than to photosynthesis at half-hourly resolution in a rice paddy, Remote Sensing of Environment, 216, 658-673, 10.1016/j.rse.2018.07.008, 2018b.

Yang, P., van der Tol, C., Campbell, P. K. E., and Middleton, E. M.: Fluorescence Correction Vegetation Index (FCVI): A physically based reflectance index to separate physiological and non-physiological information in far-red sun-induced chlorophyll fluorescence, Remote Sensing of Environment, 240, 10.1016/j.rse.2020.111676, 2020.

Yuan, W., Cai, W., Xia, J., Chen, J., Liu, S., Dong, W., Merbold, L., Law, B., Arain, A., Beringer, J., Bernhofer, C., Black, A., Blanken, P. D., Cescatti, A., Chen, Y., Francois, L., Gianelle, D., Janssens, I. A., Jung, M., Kato, T., Kiely, G., Liu, D., Marcolla, B., Montagnani, L., Raschi, A., Roupsard, O., Varlagin, A., and Wohlfahrt, G.: Global comparison of light use efficiency models for simulating terrestrial vegetation gross primary production based on the LaThuile database, Agricultural and Forest Meteorology, 192–193, 108–120, https://doi.org/10.1016/j.agrformet.2014.03.007, 2014.

Zarco-Tejada, P. J., González-Dugo, V., and Berni, J. A. J.: Fluorescence, temperature and narrow-band indices acquired from a UAV platform for water stress detection using a micro-hyperspectral imager and a thermal camera, Remote Sensing of Environment, 117, 322-337, 10.1016/j.rse.2011.10.007, 2012.

Zarco-Tejada, P. J., Morales, A., Testi, L., and Villalobos, F. J.: Spatio-temporal patterns of chlorophyll fluorescence and physiological and structural indices acquired from hyperspectral imagery as compared with carbon fluxes measured with eddy covariance, Remote Sensing of Environment, 133, 102-115, 10.1016/j.rse.2013.02.003, 2013.

Zarco-Tejada, P. J., Miller, J. R., Mohammed, G. H., Noland, T. L., and Sampson, P. H.: Estimation of chlorophyll fluorescence under natural illumination from hyperspectral data, International Journal of Applied Earth Observation and Geoinformation, 3, 7, 2001.

Zeng, Y., Badgley, G., Dechant, B., Ryu, Y., Chen, M., and Berry, J. A.: A practical approach for estimating the escape ratio of near-infrared solar-induced chlorophyll fluorescence, Remote Sensing of Environment, 232, 10.1016/j.rse.2019.05.028, 2019.

Zhang, Z., Zhang, Y., Zhang, Q., Chen, J. M., Porcar-Castell, A., Guanter, L., Wu, Y., Zhang, X., Wang, H., Ding,
D., and Li, Z.: Assessing bi-directional effects on the diurnal cycle of measured solar-induced chlorophyll fluorescence
in crop canopies, Agricultural and Forest Meteorology, 295, 10.1016/j.agrformet.2020.108147, 2020.
Zhang, Z., Zhang, Y., Zhang, Y., Gobron, N., Frankenberg, C., Wang, S., and Li, Z.: The potential of satellite FPAR
product for GPP estimation: An indirect evaluation using solar-induced chlorophyll fluorescence, Remote Sensing of
Environment, 240, 10.1016/j.rse.2020.111686, 2020.
Zhao, M., Running, S., Heinsch, F. A., and Nemani, R.: MODIS-Derived Terrestrial Primary Production, 11, 635-
660, 10.1007/978-1-4419-6749-7_28, 2010.
Zhu, X. and Liu, D.: Improving forest aboveground biomass estimation using seasonal Landsat NDVI time-series,
ISPRS Journal of Photogrammetry and Remote Sensing, 102, 222-231, 10.1016/j.isprsjprs.2014.08.014, 2015.