# Peer review of "Unveiling spatial and temporal heterogeneity of a tropical forest"

_Biogeosciences, 2021_

## Referee Comment (RC1)

**Summary of the research and my overall impression**

Merrick and coauthors present a novel dataset of remotely sensed vegetation indices (VIs) (NDVI, EVI, NIRv, NIRvrad, FCVI) from an UAS in a tropical forest canopy in Panama. They explore both spatial and temporal variability between indices and highlight potential uses for these indices at those varying scales. Specifically, the authors explore temporal correlations between GPP and VIs over the course of a day, diurnal changes in the spatial variation between Vis, and dominant spatial scales for variability in VI signals.

The paper is generally well written and structured and provides exciting insights on how VIs relate to each other. Both the dataset and the comparison are novel and within the scope of BG. Additionally, such direct comparisons between VIs are highly valuable because they provide insight in a field saturated with different VIs as to which VIs are most applicable for certain questions and specific strengths and limitations of each. The data collection approach is largely appropriate for the study, however, the temporal resolution of measurements is a major limitation. Additionally, the authors make claims about their findings in relation to SIF measurements that are not sufficiently substantiated. These major concerns are outlined with more specifics below.

Overall, this an interesting study that will be of interest to the scientific community but needs some revisions to clarify what their findings are vs. what their findings imply. Therefore, I recommend this paper be accepted with major revisions.

**Major Concerns**

- The methods section is quite dense and difficult to follow. This makes it challenging for the reader to connect measurement approach to the presented results. I recommend the authors present some sort of conceptual figure showing their measurement approach and processing. I think this will be highly beneficial, particularly for a study that explores spatial variability.

- Only one day of GPP data is available. This has led to two specific issues:
  - I am concerned about the validity of a single day's worth of GPP data. I feel as though the statistics used to partition GPP from NEE may be insufficient with only one day available. It's worth some discussion about the limitations of this approach at a minimum.
  - In section 3.1, the authors explore the diurnal trend in VI, PAR, and GPP data. They use this trend to draw conclusions over the utility of NIRvrad as a proxy for GPP. However, I do not believe one day of data is sufficient to draw such strong conclusions. Additionally, there is insufficient discussion over how potential physical (illumination, viewing direction, etc.) or environmental effects (drought, seasonality, etc.) may impact these conclusions and the limitations posed by one day of data. Finally, Figure 1 appears to show a higher correlation between GPP and PAR than between GPP and NIRvrad – therefore significantly undercutting the authors main claim in this section – that NIRvrad is an appropriate proxy for GPP over short temporal scales. To me, this section would be better off as a discussion of how NIRvrad in fact does **not** sufficiently capture diurnal variability in GPP – and moreso reflects changes in PAR. I also

recommend the authors provide a bit of additional commentary on why the other VIs show low correlations with GPP data.

- The authors repeatedly draw the conclusion that presented VI data is suitable for separating out the physiological from the structural component of the SIF signal when SIF measurements are available. However, the authors are not presenting SIF data and therefore not substantiating this claim with sufficient results or appropriate citations. Specific comments are included in specific examples. I feel that much of the SIF discussion in fact takes away from the authors main conclusions and novelty of their other results as it focuses the discussion on what they aren't doing (normalizing SIF with VI data). In particular, the majority of the introduction focuses on SIF. I recommend the authors cut down on this discussion significantly and make it more clear what conclusions they are drawing from their results vs. potential directions for future work.

**Specifics:**
- Lines 16-18: The statement 'presented here for the first time' is a bit misleading since you are not presenting these VI's for the first time, you're presenting them at this specific field site for the first time. Additionally, this opening does not make it clear the scientific question or problem you are trying to address or appropriate background information.
- Line 38: Unoccupied might be a more appropriate term, as presumably the UAS was piloted (just not with someone on board)
- Line 57: 'SIF is mechanistically linked to photosynthesis of plants, and thereby, has also been shown to be more sensitive to changes in forest canopy function and structure than RIs' – this deserves a citation. I also don't think you can say it's more sensitive to changes in forest canopy *structure* (although function yes). See the following for comparisons between SIF and VI's (among others):
  - Cheng, R., Magney, T. S., Dutta, D., Bowling, D. R., Logan, B. A., Burns, S. P., Blanken, P. D., Grossmann, K., Lopez, S., Richardson, A. D., Stutz, J., & Frankenberg, C. (2020). Decomposing reflectance spectra to track gross primary production in a subalpine evergreen forest. *Biogeosciences*, *17*(18), 4523–4544. https://doi.org/10.5194/bg-17-4523-2020
  - Magney, T. S., Bowling, D. R., Logan, B. A., Grossmann, K., Stutz, J., Blanken, P. D., Burns, S. P., Cheng, R., Garcia, M. A., Köhler, P., Lopez, S., Parazoo, N. C., Raczka, B., Schimel, D., & Frankenberg, C. (2019). Mechanistic evidence for tracking the seasonality of photosynthesis with solar-induced fluorescence. *Proceedings of the National Academy of Sciences of the United States of America*, *116*(24), 11640–11645. https://doi.org/10.1073/pnas.1900278116
  - Pierrat, Z., Nehemy, M. F., Roy, A., Magney, T., Parazoo, C., Laroque, C., Pappas, C., Sonnentag, O., Bowling, D. R., Seibt, U., Ramirez, A., Helgason, W., Barr, A., & Stutz, J. (2021). Tower-based remote sensing reveals mechanisms behind a two-phased spring transition in a mixed-species boreal forest. *Journal of Geophysical Research: Biogeosciences*. https://doi.org/10.1029/2020JG006191

- Line 87-89: It's worth mentioning which ecosystem types because this is not true across all ecosystems/some types show much better performance than others. The citations you have all have ecosystem type information.
- Lines 99-101: Again it's worth mentioning ecosystem type here (ie: specifically tropical in your case) – this doesn't necessarily apply for all ecosystems/we don't have enough studies testing this across varied vegetation cover.
- Lines 111-113: This deserves a citation (or several).
- Line 124: The introduction deserves some final statement about the broader aims of this work. What ultimate goal this information provides.
- Line 146: there's a period . typo after 12 ms
- Line 160: As mentioned above there should be additional discussion on the limitations of only one day of data.
- Line 173: I believe the original citation for NDVI is:
  - Tucker, C. J. (1979). Red and photographic infrared linear combinations for monitoring vegetation. *Remote Sensing of Environment*, *8*(2), 127–150. https://doi.org/10.1016/0034-4257(79)90013-0
- Figure 1: There appears to be some sort of accidental grid to the side of panel d?
- Lines 236-238: 'Our results demonstrate that UAS-based data are suitable for normalizing SIF at high spatial resolution in addition to recording structural heterogeneity of a tropical forest' – your results don't really demonstrate this because you don't have SIF data. Maybe if you say they have 'the potential' however I still think this distracts here from the other findings.
- Line 239: 'Because NIRv and NIRvrad use NDVI, these results also indicate that including NIR reflectance or NIR radiance is the largest contributing factor to this variability' – This is built into the definitions of NIRv and NIRvrad so I would rephrase this to reflect that.
- Lines 250-251: rephrase for clarity to 'The low variability and high means at midday of NIRv, FCVI, and NIRVrad indicate that…'
- Line 266: 'strong peak' is a bit of an overstatement, it seems much more rounded to me
- Line 277: remove 'note how'
- Lines 286-297: This discussion of SIF is much better because it acknowledges the potential, but also notes that SIF measurements are not available. This however also deserves some citations
- Line 313: Remove 'for the first time' – it's confusing as you're not presenting new indices, you're presenting new data at this particular location
- Line 317: I do not believe you can draw this conclusion with one day of data (see my major concern above)
- Lines 334-337: SIF discussion here is distracting from your main points
- Lines 345-346: You do not show that these measurements can be used to separate the components of a SIF signal and you're also not really showing how to use it as an estimate of fPAR, APAR, or GPP. Also worth noting this is for a tropical ecosystem.

---

## Author Comment (AC4)

**BG Reviewer Comments:**
**RC1**: 'Comment on bg-2021-95', Anonymous Referee #1, 04 May 2021
**Summary of the research and my overall impression**

Merrick and coauthors present a novel dataset of remotely sensed vegetation indices (VIs) (NDVI, EVI, NIRv, NIRvrad, FCVI) from an UAS in a tropical forest canopy in Panama. They explore both spatial and temporal variability between indices and highlight potential uses for these indices at those varying scales. Specifically, the authors explore temporal correlations between GPP and VIs over the course of a day, diurnal changes in the spatial variation between Vis, and dominant spatial scales for variability in VI signals.

The paper is generally well written and structured and provides exciting insights on how VIs relate to each other. Both the dataset and the comparison are novel and within the scope of BG. Additionally, such direct comparisons between VIs are highly valuable because they provide insight in a field saturated with different VIs as to which VIs are most applicable for certain questions and specific strengths and limitations of each. The data collection approach is largely appropriate for the study, however, the temporal resolution of measurements is a major limitation. Additionally, the authors make claims about their findings in relation to SIF measurements that are not sufficiently substantiated. These major concerns are outlined with more specifics below.

Overall, this an interesting study that will be of interest to the scientific community but needs some revisions to clarify what their findings are vs. what their findings imply. Therefore, I recommend this paper be accepted with major revisions.

**Major Concerns**

- The methods section is quite dense and difficult to follow. This makes it challenging for the reader to connect measurement approach to the presented results. I recommend the authors present some sort of conceptual figure showing their measurement approach and processing. I think this will be highly beneficial, particularly for a study that explores spatial variability.

    Thank you for this suggestion. We have included a methods and materials summary diagram in Section 2 as a new Figure 1 (Lines 512-521).

- Only one day of GPP data is available. This has led to two specific issues:

  o I am concerned about the validity of a single day's worth of GPP data. I feel as though the statistics used to partition GPP from NEE may be insufficient with only one day available. It's worth some discussion about the limitations of this approach at a minimum. I believe Matteo uses more data to estimate GPP but we only present one day. Ask him to explain and fill in this response?

      ▪ Thank you for pointing out the ambiguity in our description of the GPP from eddy covariance. The GPP estimates were derived from eddy covariance system data continuing several months, from which we extracted the one day of data. Unfortunately, due to a power issue, these data were not available for the first day corresponding with the hyperspectral and lidar data collection.

- In section 3.1, the authors explore the diurnal trend in VI, PAR, and GPP data. They use this trend to draw conclusions over the utility of NIRvrad as a proxy for GPP. However, I do not believe one day of data is sufficient to draw such strong conclusions. Additionally, there is insufficient discussion over how potential physical (illumination, viewing direction, etc.) or environmental effects (drought, seasonality, etc.) may impact these conclusions and the limitations posed by one day of data. Finally, Figure 1 appears to show a higher correlation between GPP and PAR than between GPP and NIRvrad – therefore significantly undercutting the authors main claim in this section – that NIRvrad is an appropriate proxy for GPP over short temporal scales. To me, this section would be better off as a discussion of how NIRvrad in fact does **not** sufficiently capture diurnal variability in GPP – and moreso reflects changes in PAR. I also recommend the authors provide a bit of additional commentary on why the other VIs show low correlations with GPP data.

  - Thank you for the suggestion for added clarity and purpose. We address the limitations of using one day of data throughout the manuscript, specifically in lines 488; 490-492, 534-538, 704-706, 735-737.  Throughout the manuscript we have made changes to carefully state that there is greater potential for NIRvrad as a proxy for GPP compared to the reflectance-based vegetation indicators (indices). The reflectance-based indicators, NDVI, EVI, NIRv, and FCVI, have been shown to trend seasonally with GPP in most biomes, but by virtue of calculating reflectances, these omit short timescale changes in incoming, scattered, and reflected radiation. NIRvrad, in contrast to reflectance-based indicators, includes the incoming, scattered, and reflected radiation in the NIR region. For this reason, recent studies (e.g. Wu et al 2020) and our study are pointing to the potential of NIRvrad to trend with GPP on short timescales through a joint relationship between NIRvrad, PAR and GPP. We have added more text in the introduction, results, discussion and conclusion, to address of this for clarity, lines 192-195, 506-509, 532-540, 866-869, and 954-955.

- The authors repeatedly draw the conclusion that presented VI data is suitable for separating out the physiological from the structural component of the SIF signal when SIF measurements are available. However, the authors are not presenting SIF data and therefore not substantiating this claim with sufficient results or appropriate citations. Specific comments are included in specific examples. I feel that much of the SIF discussion in fact takes away from the authors main conclusions and novelty of their other results as it focuses the discussion on what they aren't doing (normalizing SIF with VI data). In particular, the majority of the introduction focuses on SIF. I recommend the authors cut down on this discussion significantly and make it more clear what conclusions they are drawing from their results vs. potential directions for future work.

  - The authors appreciate these suggestions regarding the overemphasis on SIF in the discussion and introduction. Based on this thoughtful review, we have modified the manuscript extensively to focus the introduction on the quantities measured in the study and minimize the text and references to SIF and how the quantities may relate to SIF. Specifically, we removed almost all of Lines 52-101 from the original submission. We maintained mentioning SIF in the Introduction only as the studies presented compared NIRv, FCVI, or NIRvrad specifically to GPP and SIF (Lines 193-201), and in the Results and Discussion (Lines 729-731) and Conclusion (Lines 865-869 to make comparisons between measurement techniques for reflectance-based indices and SIF as well as emphasizing how this study might be relevant to SIF, which is an emerging, important potential measurement of GPP.

**Specifics:**

- Lines 16-18: The statement 'presented here for the first time' is a bit misleading since you are not presenting these VI's for the first time, you're presenting them at this specific field site for the first time. Additionally, this opening does not make it clear the scientific question or problem you are trying to address or appropriate background information.

    - o Thank you for assisting with clearer wording for this part of the abstract. We have removed the phrase "presented here for the first time" and modified the text (lines 16-23, 593-596) to clarify the purpose of the study. We see that the previous phrasing suggested the vegetation indicators were presented for the first time, when we only intended to point out these indicators specifically from UAV data are novel.

- Line 38: Unoccupied might be a more appropriate term, as presumably the UAS was piloted (just not with someone on board).

    - o We have updated to use the term 'unoccupied', as it is more appropriate (Lines 24 and 59).

- Line 57: 'SIF is mechanistically linked to photosynthesis of plants, and thereby, has also been shown to be more sensitive to changes in forest canopy function and structure than RIs' – this deserves a citation. I also don't think you can say it's more sensitive to changes in forest canopy structure (although function yes). See the following for comparisons between SIF and VI's (among others):

    - o Cheng, R., Magney, T. S., Dutta, D., Bowling, D. R., Logan, B. A., Burns, S. P., Blanken, P. D., Grossmann, K., Lopez, S., Richardson, A. D., Stutz, J., & Frankenberg, C. (2020). Decomposing reflectance spectra to track gross primary production in a subalpine evergreen forest. Biogeosciences, 17(18), 4523–4544. https://doi.org/10.5194/bg-17-4523-2020

    - o Magney, T. S., Bowling, D. R., Logan, B. A., Grossmann, K., Stutz, J., Blanken, P. D., Burns, S. P., Cheng, R., Garcia, M. A., KÓ§hler, P., Lopez, S., Parazoo, N. C., Raczka, B., Schimel, D., & Frankenberg, C. (2019). Mechanistic evidence for tracking the seasonality of photosynthesis with solar-induced fluorescence. Proceedings of the National Academy of Sciences of the United States of America, 116(24), 11640–11645. https://doi.org/10.1073/pnas.1900278116

- Pierrat, Z., Nehemy, M. F., Roy, A., Magney, T., Parazoo, C., Laroque, C., Pappas, C., Sonnentag, O., Bowling, D. R., Seibt, U., Ramirez, A., Helgason, W., Barr, A., & Stutz, J. (2021). Tower-based remote sensing reveals mechanisms behind a two-phased spring transition in a mixed-species boreal forest. Journal of Geophysical Research: Biogeosciences. https://doi.org/10.1029/2020JG006191.

- - ▪ Thank you for this comment. Based on this and the earlier suggestions, this portion of the manuscript was removed and portions of the manuscript referring to SIF significantly more focused on how the vegetation indicators measured related specifically to SIF. These references, however are valuable for our future work and are greatly appreciated.

- Line 87-89: It's worth mentioning which ecosystem types because this is not true across all ecosystems/some types show much better performance than others. The citations you have all have ecosystem type information.

  - We have updated this text to include the ecosystem or coverage of data, i.e., global, from the literature. This portion now appears in lines 82-84, but portions of the paragraph after these lines has also been updated to include more specifics (Line 85, Lines 88-92, Lines 94-98),

- Lines 99-101: Again it's worth mentioning ecosystem type here (ie: specifically tropical in your case) – this doesn't necessarily apply for all ecosystems/we don't have enough studies testing this across varied vegetation cover.

  - Thank you for pointing out this omission. We have now included text to clarify the data used in previous studies, which helps us highlight the tropical forest on which we focused (lines 94-98).

- Lines 111-113: This deserves a citation (or several).

  - Thank you for pointing out this ambiguous statement. We have removed references to using the emerging indices to potentially separate the SIF signal into physiological and physical components, as we did not test this. As a part of this process, this particular phrase was removed.

- Line 124: The introduction deserves some final statement about the broader aims of this work. What ultimate goal this information provides.

  - Thank you for this suggestion. We added a sentence at the beginning of the last paragraph of the introduction (Lines 99-104) to state the broader aims.

- Line 146: there's a period . typo after 12 ms.

  - Thank you, this error has been corrected (Line 129).

- Line 160: As mentioned above there should be additional discussion on the limitations of only one day of data.

  - Thank you for reminder here. We have addressed this in lines 488; 490-492, 534-538, 704-706, 735-737.

- Line 173: I believe the original citation for NDVI is:

- o Tucker, C. J. (1979). Red and photographic infrared linear combinations for monitoring vegetation. Remote Sensing of Environment, 8(2), 127–150. https://doi.org/10.1016/0034-4257(79)90013-0.
    - ▪ Thank you for pointing out this oversight, we have inserted this citation (Line 158).
- Figure 1: There appears to be some sort of accidental grid to the side of panel d?

    Thank you for catching our oversight. The figure has been corrected (Now Fig. 2, Line 218).

- Lines 236-238: 'Our results demonstrate that UAS-based data are suitable for normalizing SIF at high spatial resolution in addition to recording structural heterogeneity of a tropical forest' – your results don't really demonstrate this because you don't have SIF data. Maybe if you say they have 'the potential' however I still think this distracts here from the other findings.
    - o Thank you, we agree and we have removed this reference to normalizing SIF and focused this portion of the manuscript on NIRv, FCVI, and NIRvrad instead (Lines 229-234).
- Line 239: 'Because NIRv and NIRvrad use NDVI, these results also indicate that including NIR reflectance or NIR radiance is the largest contributing factor to this variability' – This is built into the definitions of NIRv and NIRvrad so I would rephrase this to reflect that.
    - o Thank you. Lines 233-234 have been updated to clarify this point.
- Lines 250-251: rephrase for clarity to 'The low variability and high means at midday of NIRv, FCVI, and NIRVrad indicate that…' T
    - o These lines, now Lines 247-250 have been revised to make this point more clearly. Thank you for suggesting a change in wording here.
- Line 266: 'strong peak' is a bit of an overstatement, it seems much more rounded to me.
    - o Thank you, we have rephrased to "distinct" to avoid overstating the shape of the peak (Line 262).
- Line 277: remove 'note how'.
    - o We have removed this part of the sentence (Line 272).
- Lines 286-297: This discussion of SIF is much better because it acknowledges the potential, but also notes that SIF measurements are not available. This however also deserves some citations.
    - o Thank you. We have included the appropriate citations for this statement in the revised version (Lines 290-298).
- Line 313: Remove 'for the first time' – it's confusing as you're not presenting new indices, you're presenting new data at this particular location.

- o Thank you, we removed this from that line (now Line 315), and created a new sentence (Lines 315-316) to clarify that we think we are the first to use such high spatial resolution data of NIRv, FCVI, and NIRvrad (from UAS). Based on this helpful suggestion, we think this more correctly asserts the claim.

- Line 317: I do not believe you can draw this conclusion with one day of data (see my major concern above).

  - o We appreciate this suggestion and we re-worded this sentence (now Lines 317-318) to discuss the potential, as well as throughout the manuscript.

- Lines 334-337: SIF discussion here is distracting from your main points.

  - o We see this now and agree. We have removed references to SIF and SIF disaggregation from the conclusions.

- Lines 345-346: You do not show that these measurements can be used to separate the components of a SIF signal and you're also not really showing how to use it as an estimate of fPAR, APAR, or GPP. Also worth noting this is for a tropical ecosystem.

  - o We have also removed these and updated this portion of the manuscript to reflect this helpful advice. Instead, we discuss the importance of future work using these vegetation indicators in tropical ecosystems and beyond to explore vegetation structure and function (Lines 337-344).

**RC2**: ['Comment on bg-2021-95'](), Anonymous Referee #2, 13 May 2021  reply

- **General Comments**

  o The authors present a very interesting and novel dataset of high-resolution vegetation indices (VI) in a tropical forest. They present correlations of the VIs to the gross primary productivity (GPP) of this forest and show how the VIs compare in capturing GPP for a given day. The authors also present a comparison of the VIs in their ability to capture structural heterogeneity of the forest. I found the study to be relevant and current given the emerging VIs used in this study. The spatial component of this study is very interesting as well. Here the authors show that NIRv and FCVI can capture more spatial heterogeneity in this forest in the reflection and absorption of radiation. My comments mostly focus on encouraging enhancement of the discussion that could provide more context for the analysis that was done and reducing the discussion of distracting concepts that were not tested. To tie the introduction and discussion to the analysis and results, the discussion and the introduction could better explain why NIRv_rad would be correlated to GPP with a clearer explanation of the GPP and NIRv (reflectance or radiance based) relationship and a reduced discussion of the role of the VIs in the SIF-GPP relationship. The paper could benefit from discussing the connections between canopy structure (height, size of tree clusters) and function (GPP) rather than the links between VIs and SIF. Below are some specific comments.

- **Specific comments**

  o The Light-Use Efficiency (LUE) model is the most widely used model to explain the relationship between GPP and vegetation indices such as NDVI as mentioned by the authors in line 42. I find the description of the LUE model to be inadequate in this paper considering it plays such a key role in understanding why vegetation indices correlate with GPP. Thinking of NIRv as an indicator of fPAR x f_esc could serve an analysis which includes observed SIF, but for the current analysis, it would be better to discuss NIRv_rad as an indicator for APAR. I would encourage the authors to present either: the equation for the LUE model with an explanation of the terms or a written description of the LUE logic and a description of its terms. Medlyn (1998) and Yuan et al. (2014) provide overviews of the LUE model and its terms. Presenting the LUE model can help readers understand exactly where vegetation indices fit in estimating GPP when one does not have SIF observations and would help clarify vague sentences like "thus a joint relationship between a remote sensing vegetation quantity, PAR, and GPP." (lines 206 – 207)

      ▪ This insight is particularly helpful to clarify our message for the readers. We have updated the manuscript to remove the emphasis on fPAR x fesc and to include information about the links to LUE (Lines 58-62). Additionally, based on this comment, others by this reviewer, and those made by other reviewers, we have significantly cut the introductory material related to the SIF~GPP~vegetation indicator descriptions and links because we did not measure SIF. We feel as if this provided a clearer background for our study focusing on traditional RS vegetation indicators and emerging indicators.

o Since the study focuses solely on vegetation indices, can the authors expand more on why near-infrared reflectance or reflected near-infrared radiation and the vegetation indices that are built from it have shown good correlations with GPP?

▪ We fine-tuned the introduction to the vegetation indicators and GPP to provide links, especially based on previous studies in Lines 62-70. We follow this portion of the manuscript with a careful description of the traditional and emerging vegetation indicators without pulling in tertiary information not related to what we are testing, such as SIF. We feel this now provides a better basis for our study.

o Making a clearer link between spatial canopy heterogeneity and GPP in the discussion can also help tie both the correlation and the power spectrum analysis together.

▪ Thank you for this suggestion. We have updated the introduction, results and discussion to include better links between canopy spatial heterogeneity and GPP Lines 38-44, 49-55, 228-234, 289-297.

o I find the discussion of SIF here to be a bit too extensive given that SIF was not actually tested. The authors have covered an important point in mentioning the use of NIRv to capture the structural component of observed SIF and it is worth mentioning in a sentence or two, but I think an analysis which is not focused on a comparison between SIF and VIs does not need to explain how VIs are related to SIF as extensively as has been done. Instead, a focus on how near-infrared reflectance of vegetation, canopy structure, and light capture/absorption is related to GPP could help address the actual comparison being made. If the authors want to focus on how NIRv can be used in the GPP-SIF relationship, then the links between NIRv, SIF, and GPP need to be discussed further to allow a reader to understand what role NIRv plays in estimating GPP through the GPP-SIF relationship. Expanding the fPAR x f_esc equation to show the full GPP equation could help in this area. However, again, since the NIRv-GPP relationship was tested, the LUE model without SIF is a better conceptual glue for this analysis.

▪ Thank you for these helpful and very detailed suggestions. Based on this reviewer's perspective, we updated the manuscript, especially the introduction, to increase the focus on NIRv, FCVI, and NIRvrad and reduce the focus on SIF. Specifically, we removed paragraph 2 from the introduction, as well as extraneous references to SIF in Paragraph 3 (Lines 52-101). We only retained a reference to SIF in regard to comparing techniques for measurement (Lines 78-81), measurements of FCVI in our study related to SIF (Lines 88-90), studies specifically comparing the vegetation indices to GPP and SIF (Line 96), and in the discussion regarding uses for emerging vegetation indices (Lines 295-296 and 316-319).

o Line 113: Can the authors expand on why NIRv needs to serve as a proxy for SIF if it can serve as a proxy for GPP and a radiance based NIRv can serve as a proxy for APAR? Using NIRv for addressing the structural component of the SIF-GPP relationship makes sense, but the utility of using NIRv as a proxy for SIF is not as clear.

- We agree that stressing the connection between NIRv, NIRvrad, and SIF takes away from the central message that these metrics from UAS provide fine-scale structural information that may help address gaps in understanding GPP. Based on this helpful suggestion, we have scaled back references to SIF, and specifically removed the references in Line 113.

- R in equation 3 and equation 4 is not explained until after equation 5. It can be clearer to explain what R represents after equation 3 and 4.

  - Apologies for this oversight. We have now corrected this omission (Lines 158-161).

- It is unclear how this analysis supports the claim at line 236 since normalizing SIF with the UAS data was not done in this study.

  - Thank you, we have removed this reference to normalizing SIF as a part of focusing the manuscript more clearly on NIRv, FCVI, and NIRvrad (Lines 230).

- Claims made at the following lines need citations: line 32 – 33, lines 56 – 57, lines 75 – 76, lines 78 – 80, lines 91 – 94

  - We added appropriate citations for lines 32-33 (now Lines 41-42). Due to modifications related to decreasing the discussions of SIF in the introduction, Lines 52-101 were removed from the manuscript. Thank you for pointing out these omissions.

- **Technical Corrections**

  - Line 49 – 50: consider changing "and questions linger about their ability to track green-up with RIs in tropical regions" to "and questions linger about the ability to track green-up with RIs in tropical regions" or "and questions linger about their ability to track green-up in tropical regions"

    - Thank you for this helpful suggestion, we have reworded according to your advice (lines 49-53.

  - Line 84: consider changing "SIF signal or used to independently as" to "SIF signal or used independently as".

    - This is a helpful suggestion, but this sentence has been removed in this revision.

Thank you for bringing these references to our attention. We have corrected this omission and included the information and appropriate references (Lines 59, 61, 68, 556-557, 671-676.

[revised manuscript text omitted]

---

## Referee Report (RR1)

As stated in my previous review, Merrick and coauthors present a novel dataset of remotely sensed vegetation indices (VIs) (NDVI, EVI, NIRv, NIRvrad, FCVI) from an UAS in a tropical forest canopy in Panama. They explore both spatial and temporal variability between indices and highlight potential uses for these indices at those varying scales. Specifically, the authors explore temporal correlations between GPP and VIs over the course of a day, diurnal changes in the spatial variation between Vis, and dominant spatial scales for variability in VI signals.

I continue to be in support of the acceptance and publication of this manuscript and am generally happy with the authors response. However, I have a few more minor comments I feel should be addressed prior to publication. They are outlined as follows:

Lines 24-25: the sentence will be more clear if rephrase it to say "…. these indices and the properties that are presumed to be measured by these indices, such as gross primary productivity (GPP) and absorbed photosynthetically active radiation (APAR)."

Line 25: 15cm and greater should just provide the maximum spatial size

Line 27: 'emerging vegetation indicators' is unclear and not specific enough – better to name them which will also help with the papers longevity when the indicators are no longer 'emerging'. This appears a few other times and I tried to catch all of them but worth doing a read through to double check.

Line 34: same comment as line 27

Line 38: the 'are' in 'are not well characterized' refers to 'spatial and temporal heterogeneity' so I believe it should be 'is not well characterized'

Line 40: change to 'forests'

Lines 59-64: This section might be clearer to read if you just replace the wording with equations then defining the acronyms

Line 70: Here you could probably replace 'tropical regions' with a statement about evergreen regions as a whole since the statement still applies and it's a little broader. Keeping it as is also works.

Line 79: SIF has not yet been defined – can remove it from this portion entirely or add a separate sentence describing what SIF is (although I think it would be better without)

Lines 83-86: This sentence is too long and convoluted. Break it up.

Line 89: change to 'demonstrated that FCVI tracked GPP…' so you keep the subject and the verb close together – it makes the sentence more straightforward

Lines 96-98: This sentence is also fairly long and convoluted. Consider replacing with an equation then a description or breaking up the sentence.

Line 145: It's unclear what the 'data corresponding to the January 30 flights' is referring to. I think there might be a typo in here.

Line 156-157: 'A summary of materials…' should go earlier in the section or can be removed entirely

Line 173: You previously say the GPP data was Jan 30 (Line 145) so one of these must be wrong

Line 173: I think it would help to reference the figures or sections where these different analysis are done immediately following the statement of what's being done. It will help provide the reader a roadmap to the different sections of analysis.

Line 204: 'a joint relationship between…' is vague. Either be more specific or remove this.

Line 216: I'm not so sure about the statement 'NIRvrad is also a more efficient measurement of GPP…' Do the authors mean to say it's more efficient than PAR? I would disagree since here it depends on the scale and the instrumentation used to take the measurements, AND the authors show a stronger agreement between GPP-PAR and NIRvrad-PAR than NIRvrad-GPP. I think the point that NIRvrad tracks APAR should be made more strongly, rather than trying to pitch NIRvrad as an alternative to PAR or APAR measurements.

Figure 2: The smoothing line is not defined. Please define it in the figure caption

Lines 237-238: It might be easier to see the comparison between the two if they are located next to each other in the figure. You could easily switch the locations of NIRv and FCVI since those are also compared with each other.

Line 245 (and the following paragraph and Figure 3): stay consistent with time notation. This line starts with 12:00 and 1330 but would be more clear if the time descriptions in the paragraph and figure are all notated the same.

Line 248: Is CV defined somewhere? Might need to be defined again or made more clear what this is referring to.

Figure 3: I think it would help to have the colors of the distributions be the same for flights at the same time but different days. For example the 15:30 flight times on Jan 30 and 31 could be colored the same to make comparison between the two easier. Further, I think it would help if the times were vertically aligned between days.

Line 296: same comment as above about 'emerging vegetation indicators'

Line 308: 'which have high reflectance in blue wavelengths compared to fully leaved crowns' deserves a citation

Line 318: remove the statement 'and the scattering of SIF photons'. Since most of this was removed from the results/discussion it seems distracting as the first sentence in the conclusions.

Line 322-323: remove 'which SIF requires' – I think the framing is better as a 'these can help inform SIF and each other' rather than pitting different RS indices against each other. Also since you're not presenting SIF data you can't make the argument NIRvrad is a more effective proxy.

Line 325: 'which may pave the way to improve our understanding of the relationship between GPP and remote sensing observations' – add a small clarification on HOW it will do this, be more specific.

---

## Author Response (AR3)

Dear Dr. Stoy and Reviewer,

Thank you for the valuable suggestions made during this final stage of our publication process of the manuscript "Unveiling spatial and temporal heterogeneity of a tropical forest canopy using high-resolution NIRv, FCVI, and NIRvrad from UAS observations" for publication in *Biogeosciences*. We appreciate your time and effort and appreciate the improvements suggested for our paper. We have incorporated changes and highlighted those within the manuscript. We also appreciate your patience while we made these changes and for your understanding related to the timeline imposed by me new agency, Naval Research Laboratory, which requires a stringent internal approval process for publishing by its scientists. Please see below for a point-by-point response to the most recent reviewer's comments.

Please accept our sincerest thanks.

On behalf of all authors,

Trina Merrick

Reviewer comments:

As stated in my previous review, Merrick and coauthors present a novel dataset of remotely sensed vegetation indices (VIs) (NDVI, EVI, NIRv, NIRvrad, FCVI) from an UAS in a tropical forest canopy in Panama. They explore both spatial and temporal variability between indices and highlight potential uses for these indices at those varying scales. Specifically, the authors explore temporal correlations between GPP and VIs over the course of a day, diurnal changes in the spatial variation between Vis, and dominant spatial scales for variability in VI signals.

I continue to be in support of the acceptance and publication of this manuscript and am generally happy with the authors response. However, I have a few more minor comments I feel should be addressed prior to publication. They are outlined as follows:

Lines 24-25: the sentence will be more clear if rephrase it to say "…. these indices and the properties that are presumed to be measured by these indices, such as gross primary productivity (GPP) and absorbed photosynthetically active radiation (APAR)."

Thank you for the suggestion. The wording has been changed.

Line 25: 15cm and greater should just provide the maximum spatial size

Thank you for pointing out this ambiguity. This adjustment has been made.

Line 27: 'emerging vegetation indicators' is unclear and not specific enough – better to name them which will also help with the papers longevity when the indicators are no longer

'emerging'. This appears a few other times and I tried to catch all of them but worth doing a read through to double check.

Line 34: same comment as line 27

We have replaced the "emerging indicators" with the names "NIRv, FCVI, and NIRvrad".

Line 38: the 'are' in 'are not well characterized' refers to 'spatial and temporal heterogeneity' so I believe it should be 'is not well characterized'

We have corrected the grammatical error in this sentence.

Line 40: change to 'forests'

We have changed from "tropical forests" to forests in this instance as you kindly suggest.

L00ines 59-64: This section might be clearer to read if you just replace the wording with equations then defining the acronyms

We have replaced this section with appropriate equations.

Line 70: Here you could probably replace 'tropical regions' with a statement about evergreen regions as a whole since the statement still applies and it's a little broader. Keeping it as is also works.

We replaced tropical with evergreen because, as you suggest, this better highlights broader applicability of the work.

Line 79: SIF has not yet been defined – can remove it from this portion entirely or add a separate sentence describing what SIF is (although I think it would be better without)

Thank you for pointing out this oversight. This reference to SIF has now been removed.

Lines 83-86: This sentence is too long and convoluted. Break it up.

We have now changed this one sentence to three shorter sentences.

Line 89: change to 'demonstrated that FCVI tracked GPP...' so you keep the subject and the verb close together – it makes the sentence more straightforward

Thank you fr this suggestion. We have rearranged this sentence for clarity.

Lines 96-98: This sentence is also fairly long and convoluted. Consider replacing with an equation then a description or breaking up the sentence.

In this case, we reworded this and the previous sentence to clarify and shorten the sentences.

Line 145: It's unclear what the 'data corresponding to the January 30 flights' is referring to. I think there might be a typo in here.

Thank you, there was a typing error in the sentence that has now been corrected.

Line 156-157: 'A summary of materials...' should go earlier in the section or can be removed entirely

We have removed this statement about the summary of materials.

Line 173: You previously say the GPP data was Jan 30 (Line 145) so one of these must be wrong

Thank you for pointing out the inconsistency. We have now corrected the data collection dates for all data to be on Jan 30 and 31, 2019, except for the EC GPP data, which was only on Jan 31, 2019.

Line 173: I think it would help to reference the figures or sections where these different analysis are done immediately following the statement of what's being done. It will help provide the reader a roadmap to the different sections of analysis.

We have added a sentence for each section and figure for results (Lines 210, 213, 221).

Line 204: 'a joint relationship between...' is vague. Either be more specific or remove this.

We have reworded this sentence to be more specific in our meaning.

Line 216: I'm not so sure about the statement 'NIRvrad is also a more efficient measurement of GPP...' Do the authors mean to say it's more efficient than PAR? I would disagree since here it depends on the scale and the instrumentation used to take the measurements, AND the authors show a stronger agreement between GPP-PAR and NIRvrad-PAR than NIRvrad-GPP. I think the point that NIRvrad tracks APAR should be made more strongly, rather than trying to pitch NIRvrad as an alternative to PAR or APAR measurements.

We have taken this valuable suggestion and rewritten much of (new) lines 252-259 to clarify the points made. We now say "The ability of NIRvrad to track APAR is notable alone. However, our evidence – albeit based on only one day of data – supports the proposed use of NIRvrad as a proxy for changes in GPP on short timescales. In future work, it may also be consequential that NIRvrad is a more practical measurement of GPP than SIF in the sense that a separate instrument to measure PAR is not needed (Wu et al., 2020; Zeng et al., 2019). Also consequential for future work, given that the relationship between NIRvrad and GPP depends on PAR, it is unclear if the association between NIRvrad and GPP would weaken during the wet season when low light or diffuse light conditions are more common (Berry and Goldsmith, 2020)."

Figure 2: The smoothing line is not defined. Please define it in the figure caption.

We added language to the caption defining the smoothing.

Lines 237-238: It might be easier to see the comparison between the two if they are located next to each other in the figure. You could easily switch the locations of NIRv and FCVI since those are also compared with each other.

The reviewer makes a very god point in this suggestion. We changed the figure to realign the distributions in one column each and assigned colors and labels to more easily facilitate comparison. However, the order the plots are presented matches the order in which NIRv, FCVI and NIRvrad were published in the literature and are thus presented in this paper. Respectfully, we feel that keeping the order consistent may prove helpful overall and prevent confusion, but hope our other changes make comparison easier for the reader.

Line 245 (and the following paragraph and Figure 3): stay consistent with time notation. This line starts with 12:00 and 1330 but would be more clear if the time descriptions in the paragraph and figure are all notated the same.

Thank you for pointing out the inconsistency in the time notation. The colon has been removed from times in (new) lines 293 – 295.

Line 248: Is CV defined somewhere? Might need to be defined again or made more clear what this is referring to.

We added the coefficient of variation (CV) explicitly in (new) line 272 as a reminder of the acronym.

Figure 3: I think it would help to have the colors of the distributions be the same for flights at the same time but different days. For example the 15:30 flight times on Jan 30 and 31 could be colored the same to make comparison between the two easier. Further, I think it would help if the times were vertically aligned between days.

Thank you for this valuable suggestion. We changed the figure to reflect these suggestions. Specifically, we realigned the distributions by time (earliest at the top to latest at the bottom), overplotted the times where data were taken at the same time, and re-assigned colors such that day and time matches are more obvious, and color coded labels for each distribution to indicate the day and time.

Line 296: same comment as above about 'emerging vegetation indicators'

We replaced this phrase with "NIRv, FCVI, and NIRvrad".

Line 308: 'which have high reflectance in blue wavelengths compared to fully leaved crowns' deserves a citation

We have added a citation to support this statement (new line 397, Bibliography line 513).

Line 318: remove the statement 'and the scattering of SIF photons'. Since most of this was removed from the results/discussion it seems distracting as the first sentence in the conclusions.

This portion has been removed and we appreciate you catching this oversight.

Line 322-323: remove 'which SIF requires' – I think the framing is better as a 'these can help inform SIF and each other' rather than pitting different RS indices against each other.

Also since you're not presenting SIF data you can't make the argument NIRvrad is a more effective proxy.

Thank you for this suggestion. That phrase has been removed and your suggested framing, 'these can help inform SIF and each other', was used to modify (new) lines 394-396 to strengthen the conclusions.

Line 325: 'which may pave the way to improve our understanding of the relationship between GPP and remote sensing observations' – add a small clarification on HOW it will do this, be more specific.

We added a sentence in (new) lines 377-379 to say, "For instance, by benchmarking changes of vegetation function and structure that underlie a GPP measurement representing the whole EC footprint, fine scale NIRv, FCVI, or NIRvrad measurements may reveal highly differential behaviors of tropical species diurnally to seasonally."